# Hippocampal place cells have goal-oriented vector fields during navigation

Jake Ormond[1✉] & John O'Keefe[1,2✉]

The hippocampal cognitive map supports navigation towards, or away from, salient locations in familiar environments[1]. Although much is known about how the hippocampus encodes location in world-centred coordinates, how it supports flexible navigation is less well understood. We recorded CA1 place cells while rats navigated to a goal on the honeycomb maze[2]. The maze tests navigation via direct and indirect paths to the goal and allows the directionality of place cells to be assessed at each choice point. Place fields showed strong directional polarization characterized by vector fields that converged to sinks distributed throughout the environment. The distribution of these 'convergence sinks' (ConSinks) was centred near the goal location and the population vector field converged on the goal, providing a strong navigational signal. Changing the goal location led to movement of ConSinks and vector fields towards the new goal. The honeycomb maze allows independent assessment of spatial representation and spatial action in place cell activity and shows how the latter relates to the former. The results suggest that the hippocampus creates a vector-based model to support flexible navigation, allowing animals to select optimal paths to destinations from any location in the environment.

Lesions of the hippocampal formation reduce an animal's ability to navigate to remembered locations, such as the escape platform in the Morris watermaze[3]. One strong candidate for underpinning navigation are the CA1 place cells that provide information about the animal's current location[4]. In open-field foraging tasks lacking a specific goal, place cells provide a non-directional measure of current position[5,6] (but see refs. [7,8]). When a goal is introduced, place cells become directional[9] and their fields move in the direction of the goal when it is moved[10,11]. Previous work has shown that cells in the hippocampal formation encode heading towards a goal and other locations in bats[12], mice[7], primates[13] and humans[14]. An important caveat in such studies is that the animal usually moves towards the goal whenever possible, precluding full assessment of the neuronal activity in non-goalward directions.

By contrast, on the honeycomb maze[2], the animal approaches the goal by a succession of binary choices between intermediate platforms, choosing the most efficient path towards the goal even when direct paths are not available (Fig. 1a,b and Extended Data Fig. 1). Notably, on each platform, the animal scans around the platform perimeter, sampling the full range of possible headings (Fig. 1c,d, Extended Data Figs. 2 and 3, and Supplementary Video 1), permitting a veridical assessment of cell firing directionality.

## Place cells are organized by ConSinks

We recorded 456 CA1 place cells (defined as having significant spatial information[15] and coherence[16]; Extended Data Fig. 4) from five rats (rat 1, 88 cells; rat 2, 94 cells; rat 3, 80 cells; rat 4, 105 cells; rat 5, 89 cells) that had successfully learned the task. Of these place cells, 142 (31.1%; $n = 21$ in rat 1, 27 in rat 2, 29 in rat 3, 30 in rat 4 and 35 in rat 5) displayed

firing patterns that were well described by vector fields converging on a location that, following vector field notation, we term a convergence sink, or ConSink (Fig. 1c–g); that is to say, these cells were tuned to the animal's egocentric heading relative to these sinks (Extended Data Fig. 5). Often, the optimal bearing to the ConSink was located in front of the animal (Fig. 1d–f), but in many instances could be to the side or behind (for example, see Fig. 1g). While ConSinks were scattered around the environment both on and off the maze, they were densest around the goal in four of the five animals (Fig. 1h and Extended Data Fig. 6a) and moved closer to the goal in the second half of the session (Extended Data Fig. 7); this clustering was not due to the positions of the place fields (Extended Data Fig. 6d–h). The population vector fields (calculated by summing platform-associated heading vectors across cells) for each animal converged to a population ConSink close to the goal (Fig. 1i and Extended Data Fig. 6b), and the average preferred directions of ConSink place cells towards their sinks were in front of the animal, −6.20°, not significantly different from 0° (Fig. 1j). ConSink tuning was stronger in every ConSink cell than allocentric head direction tuning, and removing either putative allocentric head direction candidates or cells with ConSinks beyond the maze perimeter where ConSink cells might be expected to appear most allocentric did not substantially alter the results (Extended Data Figs. 8–10).

## ConSinks point to the goal

If ConSink directional firing supports navigation to goal locations, sinks should move towards new goals after their introduction. We tested this by running the same animals in a goal shift experiment on a subsequent day (360 CA1 place cells: rat 1, 93 cells; rat 2, 96 cells;

[1]Sainsbury Wellcome Centre, London, UK. [2]Cell and Developmental Biology Department, University College London, London, UK. ✉e-mail: j.ormond@ucl.ac.uk; j.okeefe@ucl.ac.uk

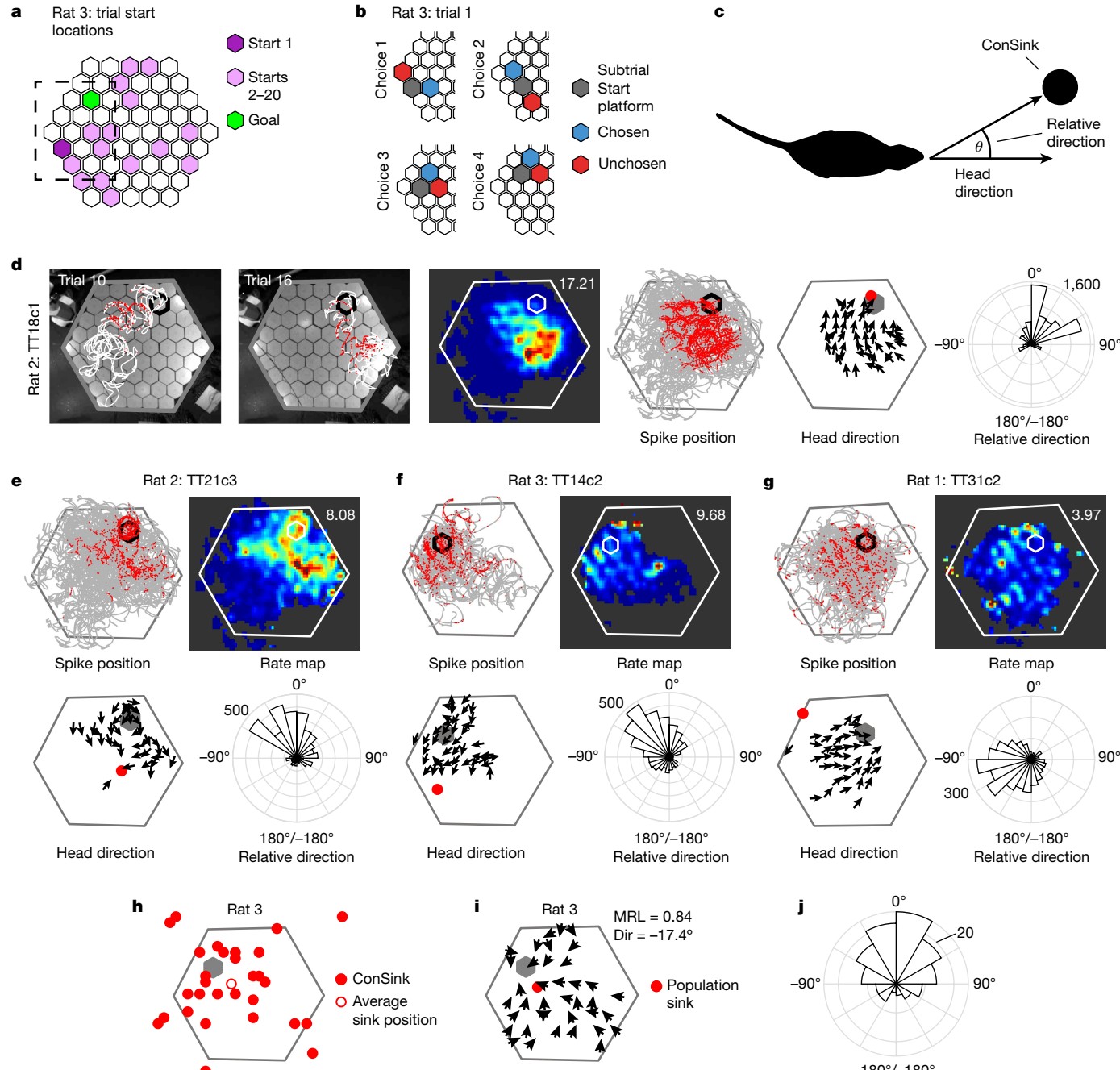

**Fig. 1 | ConSinks and vector fields organize place cell activity during navigation on the honeycomb maze. a**, Maze showing all start platforms and the goal platform for rat 3. The dashed box indicates the portion of the maze shown in **b**. **b**, Schematic of the four choices making up trial 1. The animal is confined at the 'subtrial start' until two adjacent platforms are raised and makes its choice by moving onto the 'chosen' platform. **c**, The animal's heading direction relative to a reference point (the ConSink) is calculated as the angle between the straight-ahead head direction (0°) and the direction of the point in egocentric space. **d**, Representative example of a ConSink place cell. Left two panels, paths (white) and spikes (red) fired during two individual trials of the task. The perimeter of the goal platform is shown in black. Middle two panels, place field heat map (maximal firing rate (Hz) indicated at top right) and all paths (grey) and spikes (red). Second from right, vector field depicting mean

head direction at binned spatial positions. The ConSink is depicted as a filled red circle. Right, polar plot showing the distribution of head directions relative to the ConSink. **e**–**g**, Additional examples as in **d**. **h**, ConSinks from rat 3 plotted over the maze showing a wide distribution across and off the maze (see Extended Data Fig. 6a for results from the other four animals). A grey hexagon represents the goal platform. **i**, Average vector fields for rat 3 (see Extended Data Fig. 6b for results from the other four animals). MRL, mean resultant length; Dir, mean relative direction. **j**, The mean relative directions of all significant ConSinks were non-uniformly distributed (Rayleigh test: $z = 22.19$, $P = 9.81 \times 10^{-11}$), with a mean direction of −6.20° (not significantly different from 0° (one-sample test for mean angle, $P < 0.05$); 95% confidence interval = −10.3°, 22.67°).

rat 3, 92 cells; rat 4, 79 cells): 13 trials with a familiar goal (for rats 1, 3 and 4, the same goal as in Fig. 1; for rat 2, a new goal learned before session 2; Methods) were followed, after some intermediary training, by another 13 trials to a new goal. Four of the five animals learned the new goal location within a single session (Supplementary Fig. 1), and 33% (109/331) of place cells during the original and 23% (81/347) of

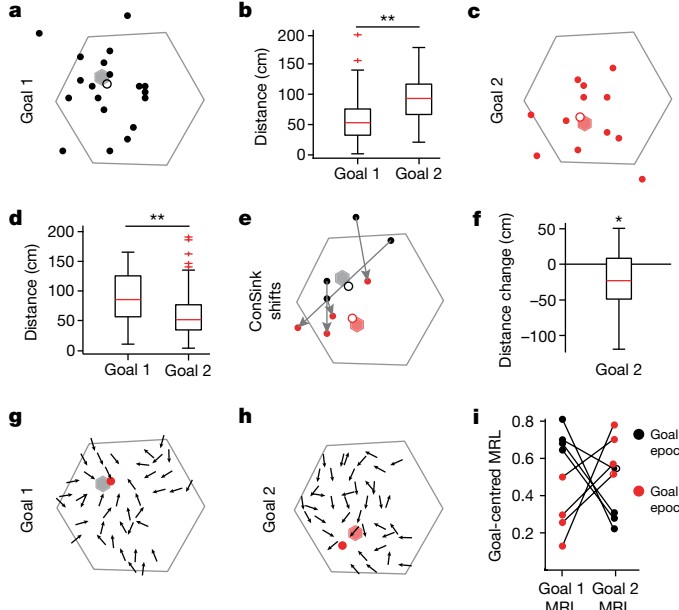

**Fig. 2 | ConSinks are under the influence of goal location. a**, Spatial distribution of ConSinks from rat 3 active only in goal 1 (grey hexagon) before the goal switch. An open circle represents the average ConSink. Results for the remaining animals are shown in Supplementary Fig. 3a–c. **b**, The ConSink population was significantly closer to goal 1 than to goal 2 before the goal switch (Wilcoxson rank-sum test, one sided: $n = 109$ cells from four animals, $z = -6.85$, $P = 3.62 \times 10^{-12}$). **c**, Spatial distribution of ConSinks active only in goal 2 (red hexagon) after the goal switch. **d**, The ConSink population was closer to goal 2 than goal 1 (Wilcoxson rank-sum test, one sided: $n = 81$ cells from four animals, $z = -4.61$, $P = 1.99 \times 10^{-6}$). **e**, Movement of ConSinks from goal 1 to goal 2 (indicated by arrows). **f**, ConSinks in **e** and Supplementary Fig. 3c were closer to the new goal after the switch (Wilcoxson signed-rank test, two sided: $n = 28$ cells from four animals, $z = -2.48$, $P = 0.013$). **g**,**h**, Population vector fields for significant ConSink cells during goal 1 (**g**) and goal 2 (**h**) for rat 3 (results for the remaining animals are shown in Supplementary Fig. 4). Population sink positions are shown as a red filled circle. **i**, MRL values taken from MRL maps (Supplementary Fig. 4) at the goals during both epochs ($n = 4$ animals). MRL values are always highest at the current goal. For all box plots, the central mark indicates the median and the bottom and top edges of the box indicate the 25th and 75th percentiles, respectively; the whiskers extend to the most extreme data points within 1.5 times the interquartile range from the bottom or top of the box, and all more extreme points are plotted individually using a plus symbol. *$P < 0.05$; **$P < 0.001$.

place cells during the shifted goal navigations had significant ConSink tuning (Supplementary Fig. 2). Before the goal shift, ConSinks were organized around the original goal, but after the switch they became organized around the new goal (Fig. 2a–d and Supplementary Fig. 3). While this reorganization primarily involved the substitution of new ConSinks for old, 28 cells with ConSinks during goal 1 continued to have ConSinks after the switch to goal 2 and the majority of these (64%) moved in the direction of the new goal (Fig. 2e,f and Supplementary Fig. 3c). Similarly, the vector fields and their associated population ConSinks moved from the original goal locations to the new goals (Fig. 2g–i and Supplementary Fig. 4). Rat 5 required multiple days to learn the new goal, and we were therefore unable to compare ConSink tuning across the two goals within a single session. Nevertheless, once goal 2 was learned (Supplementary Fig. 1f), we found that rat 5's ConSinks (Supplementary Fig. 3f; 32 of 82 place cells had significant ConSink tuning) and vector field (Supplementary Fig. 4e) were organized around the newly learned goal, as in the other animals.

Dividing the post-switch trials into two halves showed that ConSinks increased their proximity to the new goal with continued training

(Supplementary Fig. 5), similar to the movement observed during continued training to the original goal on the first day of recording (Extended Data Fig. 7). Thus, the distribution of ConSinks becomes more concentrated towards the goal as the animal repeatedly navigates to it, whether the goal is familiar or newly learned. On the other hand, while the place field centres of ConSink cells clustered around the original goal[10,17,18] (Supplementary Fig. 6a–c), they did not reorganize around the new goal, only shifting slightly in its direction. No such shift was seen in the place fields of non-ConSink cells (Supplementary Fig. 6d). We observed some remapping after the goal switch, but significantly less than that observed between the navigation task and open-field foraging (see below and Supplementary Fig. 6e,f). In the honeycomb task, reorganization of place fields appears to be a slower process than that of ConSinks, perhaps reflecting different underlying processes.

## ConSinks could support navigation

We wondered how the ConSink representation might support navigation on the honeycomb maze. Because the population vector fields showed that firing is maximal when the animal is oriented towards the goal (Fig. 1i), during unconstrained navigation the animal could simply follow the average population vector to the goal (direction G; Fig. 3a, thick red arrow). However, ConSink cells have place fields, suggesting that they not only encode relative direction (Fig. 3b), but might also encode some combination of distance and allocentric direction to the sink (Fig. 3c–e). Encoding these additional variables could support calculation of the sink positions in addition to their relative direction from the animal and allow calculation of the distance and direction vector to the goal. To test this, we used a previously published method (linear–nonlinear Poisson nested models, abbreviated LN), which, from a pool of variables of interest, identifies those that significantly improve spike prediction from a fixed mean rate model[19]. We found significant encoding of various combinations of relative direction, distance and allocentric direction to the sink in 123 of the 142 ConSink cells (Fig. 3f–j and Supplementary Figs. 7 and 8a,b), with 81 of the total (57%) encoding a combination of variables sufficient for the calculation of sink positions (that is, distance and at least one of relative or allocentric direction). The lack of significance for any variables in the remaining cells (19 of 142) could be attributed to lower firing rates (Supplementary Fig. 8c). The wide distributions of each variable (Supplementary Fig. 8d–f) indicate that the ConSink population has the information necessary to calculate the goal direction vector from any location. However, simply following the goal direction vector is not adequate when the animal is offered choices neither of which point directly to the goal. In such a case, the animal must identify the path whose heading is most similar to the goal direction by constructing and comparing the neural equivalents of the vector amplitudes in each available direction (Fig. 3a, narrow red arrows). Although we have represented the mean allocentric direction of population spiking in the vector fields (Fig. 1i), the underlying data can also be represented as a set of vectors whose lengths represent the population firing rates in the corresponding directions relative to the goal and whose average points towards the goal. We wondered whether this vector set provided enough information to allow the animal to choose between any pair of platforms as required in the navigation task. In general, the lengths of the vectors in the different directions decreased as an inverse function of the absolute angle from the average ConSink direction, forming a fantail configuration (Fig. 3k,l). Choosing the direction with the largest population vector or its equivalent, the highest population firing rate, is the correct solution to the navigation problem.

## Fewer ConSinks during foraging

Place cell firing is omnidirectional during random foraging in an open field with walls[6] (although see refs. [7,20]), but polarized or unidirectional

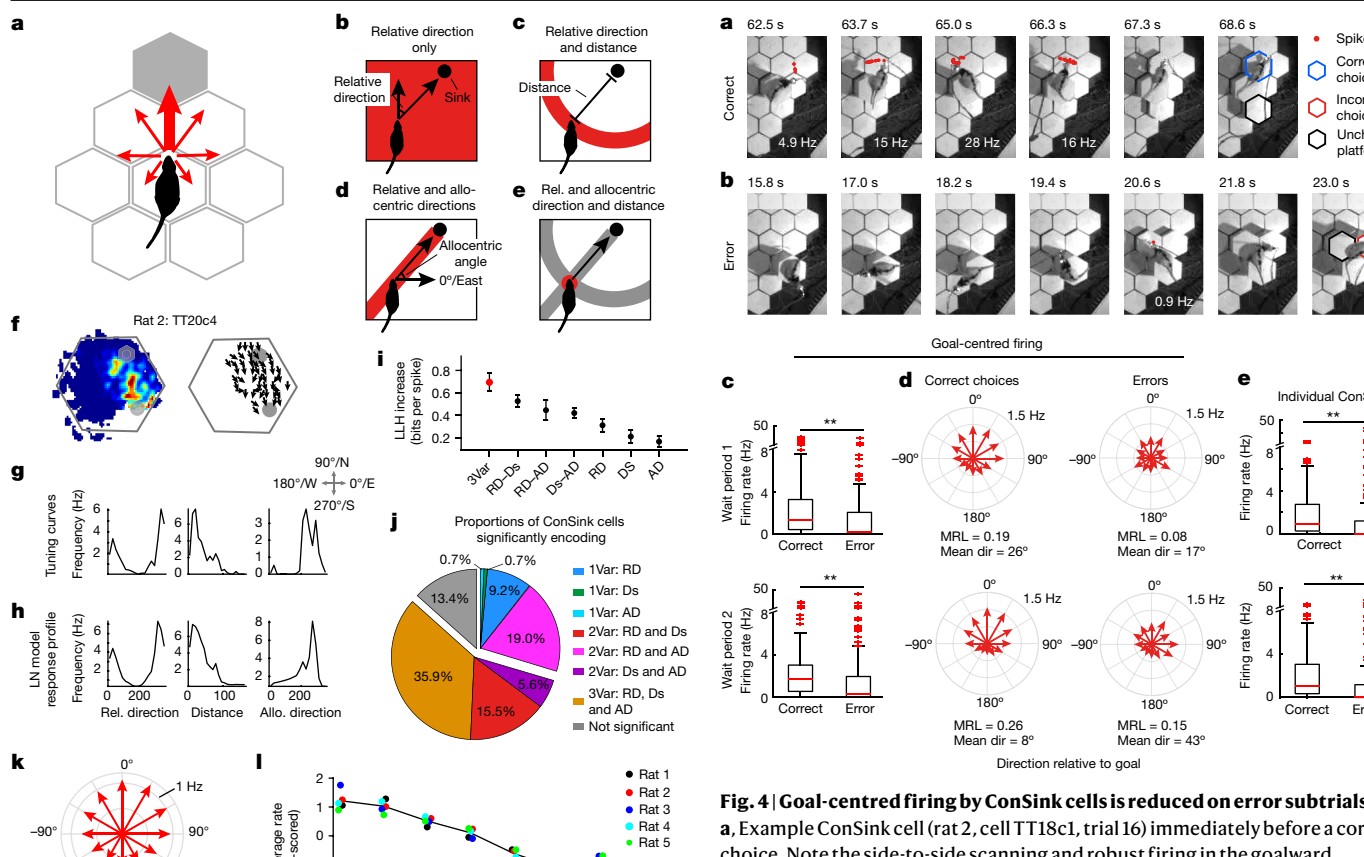

## Fig. 3 | ConSink population firing patterns contain enough spatial information to solve the honeycomb maze navigation problem.

**a**, A simple fantail model predicts that firing rate vectors will be maximum in the direction of the goal and fall off monotonically with increasing angle from the goal. **b–e**, The information necessary to construct the goal direction vector consists of all or certain combinations of the relative direction (**b**) and distance (**c**) to the ConSink and absolute direction relative to the environment (**d**), which together produce the goal direction vector (**e**). **f–h**, Typical CA1 place cell with significant information coding for all three variables with raw tuning curves (**g**) and LN model response profiles (**h**). **i**, For the cell in **f**, the combination of all three types of information provides more information than any other combination (LLH, log-likelihood; 3Var, 3 variables; RD, relative direction; Ds, distance; AD, allocentric direction). $n = 10$ repeats of the cross-validation procedure. Error bars, s.e.m. **j**, Percentages of ConSink cells encoding different combinations of the three variables in the LN model. **k**, Fantail data: population firing rate vectors across the five animals (red arrows) varying monotonically as a function of the angle between each platform direction and the population goal vector (Rayleigh test of non-uniformity of distribution, $P = 0$; mean direction = 10.9°). **l**, Population vectors for each animal conform to this model. Individual points correspond to the fantail vectors from each animal; note that positive and negative directional (for example, 30° and −30°) vectors are averaged. The black line represents the average across the five animals.

in linear environments[9,21], in virtual reality[20] and in open fields after goals are introduced[22]. We recorded the same CA1 pyramidal cells from the single-goal honeycomb task in an open-field foraging task during the same session (all platforms in the raised position). Thirteen percent of place cells (51/394) displayed significant tuning to ConSinks in the foraging task, significantly fewer than during the navigation task (Supplementary Fig. 9a–e). Only 13 cells had significant ConSinks in both tasks, and the shift in their sink locations, as well as preferred mean relative directions, across tasks indicated a complete reorganization

## Fig. 4 | Goal-centred firing by ConSink cells is reduced on error subtrials.

**a**, Example ConSink cell (rat 2, cell TT18c1, trial 16) immediately before a correct choice. Note the side-to-side scanning and robust firing in the goalward direction before the move onto the correct platform (goal beyond the top of the frame). **b**, Same cell as in **a** before an incorrect choice. **c**, Firing rates in the goalward direction were reduced on error trials, during both wait period 1 (the 4-s period before raising the choice platforms; top) and wait period 2 (the 4-s period before movement onto the chosen platform; bottom). Wilcoxson signed-rank test, two sided: $n = 142$ cells from five rats, $z = 5.61$, $P = 1.97 \times 10^{-8}$ (wait period 1); $z = 4.87$, $P = 1.14 \times 10^{-6}$ (wait period 2). **d**, Fantail plots of firing rates in directions relative to goal during correct choices have canonical forms, as in Fig. 3k, and peaks much closer to 0° than during incorrect choices. **e**, The tuning of ConSink cells to their individual sink positions is also disrupted on incorrect choices. Wilcoxson signed-rank test, one sided: $n = 142$ cells from five rats, $z = 3.27$, $P = 5.46 \times 10^{-4}$ (wait period 1); $z = 3.88$, $P = 5.17 \times 10^{-5}$ (wait period 2). For box plots in **c** and **e**, the central mark indicates the median and the bottom and top edges of the box indicate the 25th and 75th percentiles, respectively; the whiskers extend to the most extreme data points within 1.5 times the interquartile range from the bottom or top of the box, and all more extreme points are plotted individually using a plus symbol. **P < 0.001.

of the hippocampal representation (Supplementary Fig. 9f–h). Unlike during navigation, the distribution of sinks was not denser around the navigation task goal location than expected by chance (Supplementary Fig. 9i and Extended Data Fig. 6a). The existence of multiple random food sources in the foraging task is probably responsible for the remapping, although we cannot rule out the possibility that the difference might be due to the difference in the structures of the two maze configurations.

## Separating representation from action

Because raising of the next choice platforms is delayed after each choice, the honeycomb maze presents an opportunity to observe the animal's spatial representation before it chooses its next trajectory or knows what choices will be offered. We examined two time periods: wait period 1, which started after the animal's choice (Fig. 1b) but before the next choice platforms had started to rise (duration of 4 s), and wait

period 2, which was the 4-s window leading up to the animal's next choice (defined as the moment when the animal's torso moves onto the chosen platform). During the wait periods, the rat systematically scans the environment (Fig. 4a,b), providing an opportunity to sample the different branches of the fantail. We found that, before correct choices, the animal's behaviour on average displayed an even sampling of all directions relative to the goal, but before error choices the distribution of behavioural orientations was skewed away from the goal (Supplementary Fig. 10). The firing rates of ConSink cells were significantly higher in correct choices both in the goal direction (Fig. 4c,d) and in the preferred directions relative to the individual ConSinks (Figs. 1 and 4e). The corresponding fantails were appropriately peaked towards the goal before correct choices during both periods but incorrectly rotated away from goal before incorrect ones. Thus, it appears that the fantail distributions relative to the goal predict subsequent behavioural choice even before any knowledge of upcoming choices.

## Discussion

During navigation on the honeycomb maze, the firing patterns of a subset of CA1 hippocampal place cells are organized as vector fields oriented around a set of featureless environmental locations called ConSinks. While the total population of CA1 pyramidal place cells provides information about the animal's current location, the subpopulation of ConSink place cells contains all the information necessary for successful, flexible navigation in a familiar environment: allocentric information about the animal's location in the environment (place coding), both allocentric and egocentric information about distance, location and heading to the ConSinks, and, at a population level, fantails describing the relative goalward value of different directions from any given location. While the ConSinks are distributed throughout the environment, they are concentrated around and centred near the goal, providing clear evidence for an effect of learning on ConSink location. This is further supported by the appearance of new sinks, and the disappearance or rearrangement of existing sinks, around a new goal after a goal switch, as well as the continued movement of the sinks towards the goal during continued performance of the task within a single day. Finally, we find that firing towards the goal or the individual sinks is reduced and the fantail pattern is altered on error trials. Thus, ConSink cells exhibit properties that could support navigation to the goal.

We suggest how the hippocampus could solve the honeycomb task using ConSink cells. When the direct route to the goal is available, because the ConSinks are densest around the goal and, on average, the ConSink cells fire when their associated sinks are directly in front of the animal, the population firing rate is highest when the animal is oriented towards the goal. Thus, the average vector field points towards the goal and signals its distance. When the direct route to the goal is not available, the population rate falls monotonically with deviation from the direct goal heading, and the animal simply needs to move in the direction of highest firing rate afforded by the choices available, essentially comparing the lengths of the available branches of the fantail.

On erroneous choices, even before the new choice platforms rise, firing rates are reduced and the fantails deviate from the canonical form. The prechoice behaviour on these trials is also disrupted, with the animal spending more time looking away from the goal than towards it as though its attention had been attracted elsewhere. These behaviours both recall vicarious trial-and-error behaviour[23,24], recently explored in the context of predictive place cell firing at the choice point[25]. The honeycomb maze reveals the two-dimensional nature of the prechoice 'subjunctive' representation.

Cells similar to ConSink cells have been reported in mouse open-field foraging[7] and in humans performing a multiple object-in-location virtual reality test[14]. In the present experiment, these cells represented only 13% of cells during a foraging task, fewer than half of the 31% seen during the navigation task, and, further, were not clustered around the concurrent navigation task goal. While previous studies have reported the orientation of hippocampal formation cell firing relative to goals[12], objects[26], the centre of the environment[26,27] (although egocentric coding to environmental boundaries can appear as coding to the centre[28]) or random points scattered around the environment[7], the current work identifies a set of featureless locations dotted around the environment but organized around the goal, which they move closer to as learning proceeds. It must be left to future studies to determine how these ConSinks are created and manipulated, to elucidate the properties of the underlying reinforcement mechanism and to determine whether ConSink cells represent a distinct cell type or whether ConSink tuning is simply a property place cells can turn on and off. Examination of error choices suggests that selection of the correct choice platform cannot occur if the animal does not first activate an accurate representation of the hierarchy of choices and their valences. The process mediating the translation between spatial representation and spatial action must await further experimentation.

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

# Methods

## Subjects and surgical procedures

Subjects were five male Lister hooded rats purchased from Charles River Laboratories and aged between 9 and 12 months at the time of electrophysiological recordings. Animals were food deprived to ~85–90% of their baseline weight. All animal experiments were carried out in accordance with UK Home Office regulations (UK Animals Scientific Procedures Act of 1986; project license PPL PD8CBD97C). Study protocols were in accordance with the terms of the project license, which was reviewed by the Animal Welfare and Ethical Review Board at University College London. No statistical methods were used to pre-determine sample size. Our study did not use separate groups, so neither randomization nor blinding was used.

## Surgery and recording

Rats were anaesthetized with 0.5–1.5% isoflurane. Craniotomies were made bilaterally over the dorsal hippocampus (4.2 mm posterior to the bregma, ±3 mm lateral to the midline). The electrode array, containing 32 tetrodes (16 tetrodes per hemisphere), was implanted onto the surface of the cortex, and electrodes were turned 750 µm into the brain. One bone screw attached to the skull over the frontal cortex served as ground and reference. Tetrodes (nichrome, ¼ Hard Pac coating, 0.0005 inches in diameter; Kanthal, PF000591) were gold plated to <150 kΩ before implantation. Tetrodes were lowered to the dorsal CA1 over 2 weeks, and rats continued to run daily training sessions on the maze. Data were acquired using an Intan RHD USB interface board and RHD headstages.

## Maze

The honeycomb maze consists of 61 tessellated hexagonal platforms (11.5 cm each side) in an overall hexagonal configuration (total maze width of ~200 cm). Each platform can be raised or lowered independently on a linear actuator, with the raised position ~30 cm higher than the lowered position. Platforms were controlled with digital pulses generated in custom-written software in LabView. The presence of an animal on a given platform was detected using load cells (RobotShop, RB-Phi-117) on which the platforms were affixed. The load cell signal was amplified with a custom-made circuit and input into our custom LabView software.

Animals were initially trained to consume the food reward (honey-flavoured corn flakes) in their home cage. Once they were consuming the food in their home cage, they were brought onto the maze, placed on the reward platform and given a food reward. Once they were consuming the food on the reward platform without hesitation (after 1 or 2 days), we began to run task trials, initially running small numbers of trials and increasing the number gradually in preparation for the recording sessions. The task was run as follows. A list of 13 start platforms was created in MATLAB by randomly choosing a single platform from each of 13 maze subsections. The first start platform was raised, the animal was manually placed on it and the trial was started from the custom LabView software by the experimenter. Two adjacent platforms were then pseudo-randomly selected by the software with two stipulations: first, that at least one of the platforms provided a position closer to the goal than the animal's current position and, second, that previously unused platforms were selected from first, as long as the first stipulation could be met. The animal's choice was registered once the load cell system had registered its presence on one of the two choice platforms for a continuous 5 s. This triggered the lowering of the two other platforms, and, after a delay of 4–10 s, another choice sequence commenced. The choice was registered as correct if the animal chose the platform closer to the goal. In some choices, the two choice platforms were the same distance from the goal; these choices were not included in the analysis of behavioural performance. Once the animal reached the goal platform, a food reward (honey-flavoured corn flakes) in a small metal bowl was placed next to the animal after a short delay. Once the animal had finished consuming the reward, the experimenter placed the animal on a holding pedestal next to the maze. Every four trials, the maze was wiped down with 70% ethanol and rotated 30° (alternating between clockwise and anti-clockwise rotations) on a bushing located under the maze to prevent the animal from following scent trails to the goal.

The animal's ability to correctly navigate the maze was confirmed using the binomial test[29] to determine whether the number of correct choices was greater than would be expected by chance given a 0.5 probability of correct choices.

In the first recording sessions (Figs. 1, 3 and 4), animals ran 13 trials of the task, followed by open-field foraging and then a second set of task trials that varied in number among the five animals (13 additional trials for rats 1, 4 and 5 and 7 additional trials for rats 2 and 3). In the second recording sessions (Fig. 2), animals ran 13 trials to goal 1 followed by 13 trials to goal 2; there was a set of 'learning' trials interleaved between the two sets, detailed below in the 'Goal switch training' section.

## Spike sorting

Spikes were automatically sorted using KiloSort[30] followed by manual curation using Phy[31], which consisted mainly of merging and deleting clusters, using autocorrelations and cross-correlations as a guide. Cells with greater than 1% of spikes within the first 2 ms of spike autocorrelation were excluded from further analysis. Cells were classified as pyramidal or interneuron cells or were left unclassified (excluded from further analysis) on the basis of spike width, mean rate, burst index ($n$ spikes from 0–10 ms of the autocorrelation/$n$ spikes from 40–50 ms of the autocorrelation) and oscillation score[32] using a principal-component analysis. Principal components were calculated from the four variables, and the first two principal components were plotted as a scatterplot. Pyramidal cells tended to cluster together, while interneurons were scattered outside the main cluster; the experimenter selected the cells within the cluster by manually drawing a boundary, followed by visual verification of the waveforms.

## Video tracking

Video was recorded in custom LabView software using a monochrome USB camera at a frame rate of ~25 frames per second (Imaging Source). Tracking was performed offline using DeepLabCut[33]. In DeepLabCut, we trained the network to identify two infrared LEDs positioned on top of the animal's implant, as well as dark fur patches on the shoulders and back. The animal's head position was taken as the midpoint between the two LEDs, and its head angle was the angle of the line between the LEDs.

## Histology

Marking lesions were made using 20 µA of anodal current for 10 s. Animals were transcardially perfused with PBS followed by 4% paraformaldehyde (PFA), brains were cryoprotected in 30% sucrose in 4% PFA, and frozen slices of 40 µm were cut. Slices were stained with Cresyl Violet.

## Preprocessing of spike data

Hippocampal place cell data are typically velocity thresholded because place cells lose some place tuning when the animal stops moving. In our task, because the animal was not able to move freely around the maze, this seemed an inappropriate approach. Instead, we focused on excluding sharp wave ripple events by using three criteria that had to be met simultaneously: (1) theta power (6–12 Hz) below the mean; (2) population firing rate 2 s.d. above the mean; and (3) ripple power (100–250 Hz) 2 s.d. above the mean. Spectral power was computed by taking the absolute value of the output of the continuous one-dimensional wavelet transform (MATLAB function 'cwt'). Data were excluded only if all three criteria were met for a minimum duration of 50 ms. Spike data from a given cell were only used for analysis during any of the

conditions (honeycomb task, forage, goal 1, goal 2) if at least 500 spikes were fired in the relevant condition after this exclusion.

### Relative direction analysis

The field of view was tiled with potential ConSinks, arranged along the $x$ and $y$ axes at ~7-cm intervals (34 × 29 total positions). The head directions relative to each potential sink were then calculated for each spike by subtracting the angle of the vector from the animal's position to the sink location from the animal's allocentric head direction. Thus, if the animal was facing the sink, these two directions were equal and the relative direction was 0°. If the animal was facing in the opposite direction, the relative direction was 180°. The convention used in this paper is that positive relative directions indicate that the animal's head direction was to the left of a line from the animal to the sink (that is, the sink is to the animal's right) and negative directions indicate a rightward head direction relative to the sink. For a given cell, a binned distribution of relative directions could then be calculated (24 bins spanning −180° to 180°).

This distribution was then corrected for uneven sampling of relative directions by the animal. Because of the potential for differences in sampling of relative direction across the maze, we calculated control distributions of relative direction at each platform (61 platforms in total) using all video frames in which the animal occupied a given platform (an animal was deemed to be occupying a platform if its torso was within the platform's perimeter). For each cell, the distributions were scaled according to the number of spikes the cell fired on each platform. After scaling, the distributions were summed across platforms. Finally, the cell's relative direction distribution was divided by the summed control distribution, providing a corrected distribution taking into account any uneven sampling of relative direction by the animal across positions at which the cell fired spikes. From this corrected distribution, using the CircStat toolbox[34], we computed the Rayleigh test for non-uniformity of circular data (all ConSink cells were significantly non-uniform) and calculated the mean direction and the MRL. Thus, each cell had an MRL value associated with each potential sink; the candidate sink was taken as the potential sink with the highest MRL. Note that the same correction was also performed for calculation of allocentric head direction tuning.

To test whether a cell was significantly tuned to direction relative to the candidate sink, we shuffled the cell's head directions such that the head directions were no longer associated with actual positions on the maze. Distributions were calculated as above, yielding MRL values for each $xy$ position. For each of the 1,000 shuffles, the maximal MRL value across all $xy$ positions was used to make a distribution of MRL values. A cell was deemed to be significantly modulated by relative direction if its MRL was greater than the 95th percentile of the control distribution (Extended Data Fig. 5).

To account for disruption of the temporal structure of spiking caused by our shuffling procedure, we performed a second test in which we shifted the spike trains in time (minimum shift of 60 s), recalculating the strength of tuning to the sink position using the shifted positional and head direction values. A control distribution of MRL values was created from 1,000 repeats of the shifts, and cells were discarded if their real MRL was less than the 95th percentile of this control distribution.

Place cells frequently fired in bursts as the animal scanned the environment, causing a smearing of head direction that could potentially lead to false negatives in our search for ConSink cells. Thus, we repeated our search for ConSink cells using only bursts and averaged relative direction and position within each burst to eliminate the smearing effect. Bursts were defined as trains of at least ten spikes fired with interspike intervals of less than or equal to 0.25 s. If two bursts were separated in time by less than 0.5 s, they were combined. If a cell was significant in both analyses (15/77 cells), only the data from the burst analysis, which produced the greatest tuning in all cases, were carried forward into subsequent analyses. In these 15 cells, we confirmed that

both analyses identified the same ConSink positions (distance between ConSinks in the same cells = 10.3 cm and in different cells = 85.2 cm, $P < 0.001$; median difference in preferred relative direction, in the same cells = 3.3° and in different cells = 74.9°, $P < 0.001$).

To make the vector fields for individual ConSink cells (for example, Fig. 1d–g), the field of view was binned into 20 × 16 spatial bins and a mean head direction value was calculated for each bin with more than 20 spikes.

To confirm the validity of our methodology, we recalculated sink positions and preferred directions using a downsampling method (Extended Data Fig. 5d–i). For each cell on each platform, the spikes were binned according to allocentric head direction (24 bins of 15° in width). Spikes within each bin were then downsampled according to the total directional occupancy in that bin; that is, if the animal spent relatively more time facing a particular direction, the spikes fired in that direction were downsampled by a proportionate amount. For a given cell, spikes were then summed across platforms and the sink position, preferred relative direction and strength of tuning (MRL) were calculated. This was repeated for each cell 1,000 times, and the mean values were calculated.

For population vector fields, bins instead corresponded to maze platforms. For each cell on each platform, we calculated the cell's mean allocentric head direction. We then created a unit vector from this head direction value and scaled it by multiplying it by both its platform-associated mean firing rate and MRL. Finally, these scaled vectors were summed across all ConSink cells that fired spikes on the platform, and a direction was calculated from the resulting vector. The population sink position was calculated using the same search across $xy$ positions as for individual cells, converting each platform-associated head direction to a relative direction whose contribution was scaled according to the length of its associated vector and calculating the MRL, taking as the sink the position with the maximal MRL value.

### Behavioural analysis

To determine what platform an animal was on for post hoc analysis, we tracked the position of the animal's torso using a dark fur patch behind the shoulders in DeepLabCut[33]. The animal was considered to be on a particular platform if the torso position was within the platform's perimeter. For the analysis of correct and error choices (Fig. 4), wait period 2 was defined in relation to the time when the animal's torso moved onto the chosen platform and was taken as the 4-s window starting 5 s before this transition; the 1-s gap between the end of wait period 2 and the transition to the new platform ensured no contamination of wait period 2 by the transition itself. Wait period 1 was defined as the time after the previous subtrial start when the unchosen platforms had lowered and before the next choice platforms started to be raised.

### Goal switch training

In the goal switch trials, all animals ran 13 trials to goal 1. Once these trials were completed, it was necessary to teach the animals that the goal position had switched. To do this, we ran a number of 'easy' trials to goal 2. These easy trials were characterized by choice platforms that all led the animal closer to the new goal, such that the animal would arrive at the new goal through no real choice of its own. Once the animal arrived at the new goal, it was rewarded as normal. These trials were interleaved with easy unrewarded trials to goal 1.

The training sequences for each animal were as follows. Rat 1 ran four easy trials to goal 2, followed by a normal trial that it was not able to complete successfully. It then ran two easy unrewarded trials to goal 1, followed by a single easy rewarded trial to goal 2. It subsequently ran 13 normal trials to goal 2, all of which are included in the presented analysis.

The data presented for rat 2 are from the second goal switch session. The first goal switch session was not completed successfully owing to this rat's inability to learn the new goal location. We subsequently ran brief sessions across 6 days of 1–4 trials to this new goal location in

which the rat demonstrated clear learning of this new goal location. We then ran a second goal switch session, using goal 2 from the failed goal switch session as goal 1. After running 13 trials to goal 1, we switched the goal and the animal ran three easy rewarded trials. He subsequently ran 15 normal trials to goal 2. The final 13 of these trials were used in all analyses except for the analysis examining movement of the sinks across the two halves of the goal 2 epoch (Supplementary Fig. 5), which used the first 13 trials.

Rat 3 ran 13 trials to goal 1, followed by 8 easy trials alternating between rewarded trials to goal 2 and unrewarded trials to goal 1. This was followed by 17 normal trials to goal 2. The final 13 of these trials were used in all analyses except for the analysis examining movement of the sinks across the two halves of the goal 2 epoch (Supplementary Fig. 5), which used the first 13 trials.

Rat 4 ran 13 trials to goal 1, followed by 6 easy trials alternating between unrewarded trials to goal 1 and rewarded trials to goal 2 and 3 additional easy rewarded trials to goal 2. This was followed by 21 normal trials to goal 2. The final 13 of these trials were used in all analyses except for the analysis examining movement of the sinks across the two halves of the goal 2 epoch (Supplementary Fig. 5), which used the first 13 trials.

Rat 5 ran 13 trials to goal 1, followed by 8 easy trials alternating between unrewarded trials to goal 1 and rewarded trials to goal 2. This was followed by normal trials to goal 2, but the animal persisted in going to goal 1. A few attempts were made to 'retrain' the animal to goal 2 with additional easy trials followed by normal trials, but the animal stopped running the task before running the necessary number of trials to goal 2. This animal was therefore excluded from the main analysis. It was trained to goal 2 over several additional sessions, and data from the final session are presented in Supplementary Figs. 1, 3 and 4.

### LN analysis

To determine the dependence of ConSink cell spiking on distance and direction to the sink, we used a technique developed to identify mixed selectivity in individual neurons by quantifying the dependence of spiking on all possible combinations of a set of variables[35]. A model, corresponding to a particular combination of variables, is trained by optimizing a set of parameters that convert animal state vectors corresponding to the variables of interest into firing rates, which are estimated as an exponential function of the sum of each variable value projected onto the set of parameters. The analysis uses tenfold cross-validation, splitting the data into ten equally sized partitions, training the model using nine of the partitions and testing on the held-out partition such that each partition is tested once. The LLH increase in spike prediction relative to a mean firing rate model is calculated for each model, and the simplest model (that is, the one with the fewest number of variables) that produces a significant increase relative to the mean firing rate model, as well as, in the case of multivariable models, a significant increase over any simpler models (that is, models with fewer variables) is selected as the model that best describes the neuron's tuning. The significance is assessed using Wilcoxon signed-rank tests comparing the LLH increases for each test partition across the relevant models. We adapted the model to use three variables: (1) relative direction to the sink (RD), (2) distance to the sink (Ds) and (3) direction from the animal's position to the sink (AD); this produced seven possible models (that is, a three-variable model, three two-variable models and three single-variable models). Firing rate and animal state vectors were constructed using 100-ms windows. Relative direction and direction from position were binned using 18 bins spanning −180° to 180° and 0° to 360°, respectively. Distance to sink was binned using 20 bins from 0 cm to the animal's maximum distance from the sink.

### Fantail plots

To calculate the fantail plot (Fig. 3k) showing the population firing rates in the range of head directions relative to the goal, spikes within animals were combined across all ConSink cells. For each spike, the animal's head direction relative to the goal was calculated in the same way as for head direction relative to the ConSink (see 'Relative direction analysis' above). Spikes were then separated according to the platform occupied by the animal during spiking, and, for each platform, its associated spikes were binned according to relative direction to goal (30°-wide bins). Similarly, for each platform, the total amount of time that the animal spent within each relative direction bin was determined. Finally, the spike counts in each bin were divided by the total time (in seconds) spent in each bin, to produce firing rates in each bin. These were then averaged across all platforms and divided by the total number of ConSink cells to generate a per-cell firing rate in binned direction relative to the goal; these values, $z$-scored within animals, are shown in Fig. 3i and are averaged across animals in Fig. 3k.

### Remapping analysis

To assess remapping between the honeycomb task and open-field foraging, we created rate maps for all cells by partitioning the field of view into 1,280 bins (40 bins in the $x$ direction, 32 bins in the $y$ direction). To establish baselines for cell stability, to which we could compare our remapping data to assess significance, we created rate maps corresponding to the first and second halves of the task and open-field foraging epochs; specifically, for each spatial bin, we calculated the total occupancy (in seconds) and placed the data corresponding to the first and second halves in the corresponding rate maps. Population vector correlations were performed by constructing vectors for each bin using the firing rates of each cell at that bin and then calculating Pearson's linear correlation coefficient for the two vectors being compared. Similarly, place field correlations were performed by linearizing the two-dimensional rate maps for a given cell and calculating the correlation between the two vectors.

### Place field centres

Place fields centres were taken as the centre of mass of the cell's rate map.

### Statistical procedures

All statistical tests were two sided and non-parametric unless stated otherwise. In box plots, the central mark indicates the median and the bottom and top edges of the box indicate the 25th and 75th percentiles, respectively. The whiskers extend to the most extreme data points within 1.5 times the interquartile range from the bottom or top of the box, and all more extreme points are plotted individually using a plus symbol.

### Reporting summary

Further information on research design is available in the Nature Research Reporting Summary linked to this paper.

## Data availability

The datasets generated during the current study are available from the corresponding authors on reasonable request. Source data are provided with this paper.

## Code availability

The custom MATLAB code used for all analyses can be accessed at GitHub (https://github.com/jakeormond/vectorFields).

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

**Acknowledgements** We thank D. Halpin and S. Townsend for building the honeycomb maze and, with R. Barrett, for help prototyping the platforms and modifying the maze; P. Sienkiewicz for designing and building the electronics; G. McPhillips and M. Stopps for consulting on the load cell circuitry; A. Radulovic for help with data collection; B. Greenaway and J. Broni-Tabi for histology; and M. Bauza and members of the O'Keefe laboratory for helpful discussions regarding the results and manuscript. This work was supported by the Sainsbury Wellcome Centre Core Grant from the Gatsby Charitable Foundation and Wellcome Trust (090843/F/09/Z) and a Wellcome Trust Principal Research Fellowship (Wt203020/z/16/z) to J.O'K.

**Author contributions** J.O. and J.O'K. designed the experiments. J.O. collected the data. J.O. and J.O'K. analysed the data and wrote the manuscript.

**Competing interests** The authors declare no competing interests.

**Additional information**
**Correspondence and requests for materials** should be addressed to Jake Ormond or John O'Keefe.

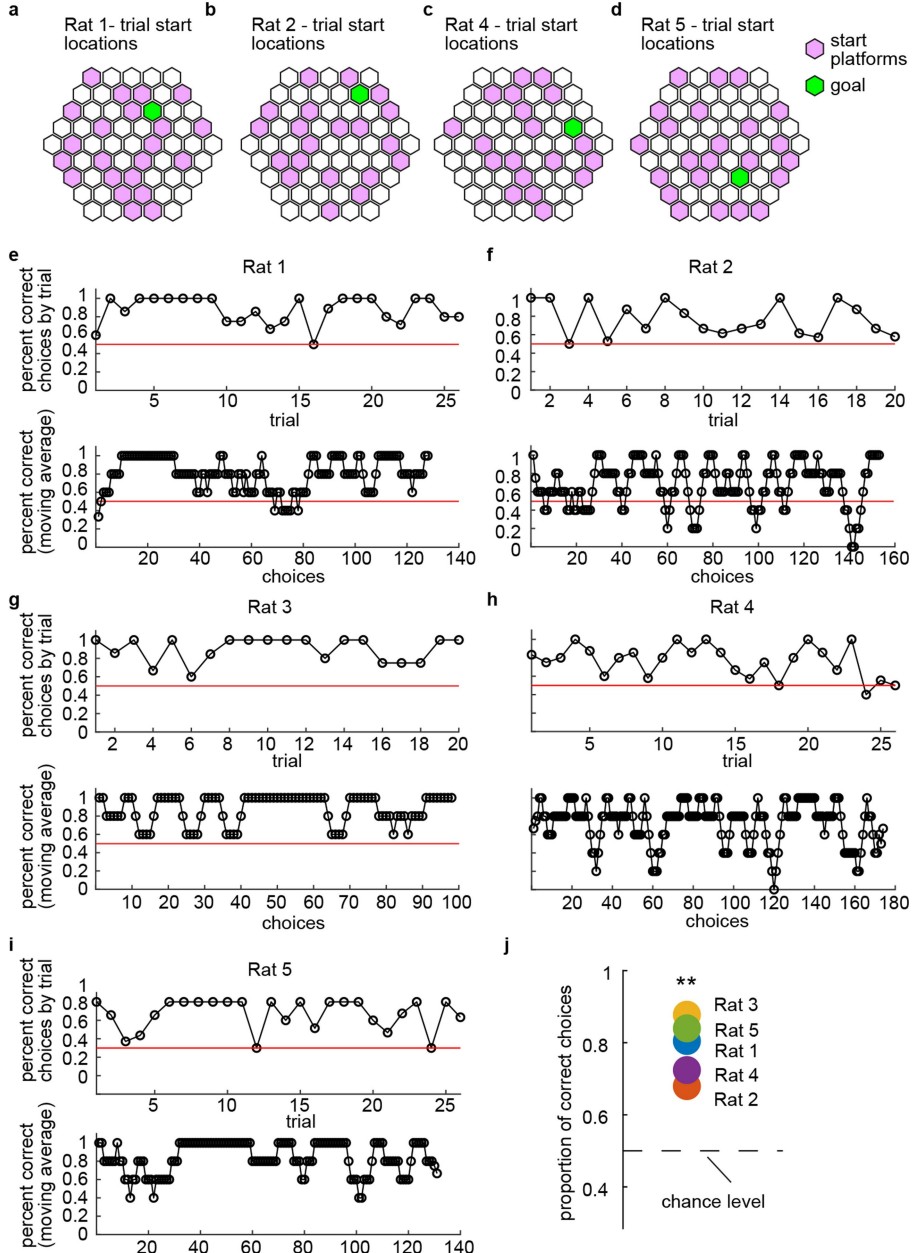

**Extended Data Fig. 1 | Behavioural Summary. a**, Schematic of the honeycomb maze showing all start platforms and the goal platform from Rat 1's session. **b-d**, Same as is in (**a**) but for Rats 2, 4, and 5, respectively. **e**, Top panel, Percentage of choices that were correct averaged by trial for Rat 1. Bottom panel, Running average of correct choices (every 5 consecutive choices). **f-i**, As in (**e**) but for Rats 2 to 5, respectively. **j**, The total proportion of correct choices for each rat. Each rat made significantly more correct than incorrect choices; two-sided binomial test within each animal, p = 5.17 x 10$^{-5}$, 1.03 x 10$^{-5}$, 5.96 x 10$^{-15}$, 1.32 x 10$^{-4}$, 9.16 x 10$^{-16}$ for rats 1–5, respectively.

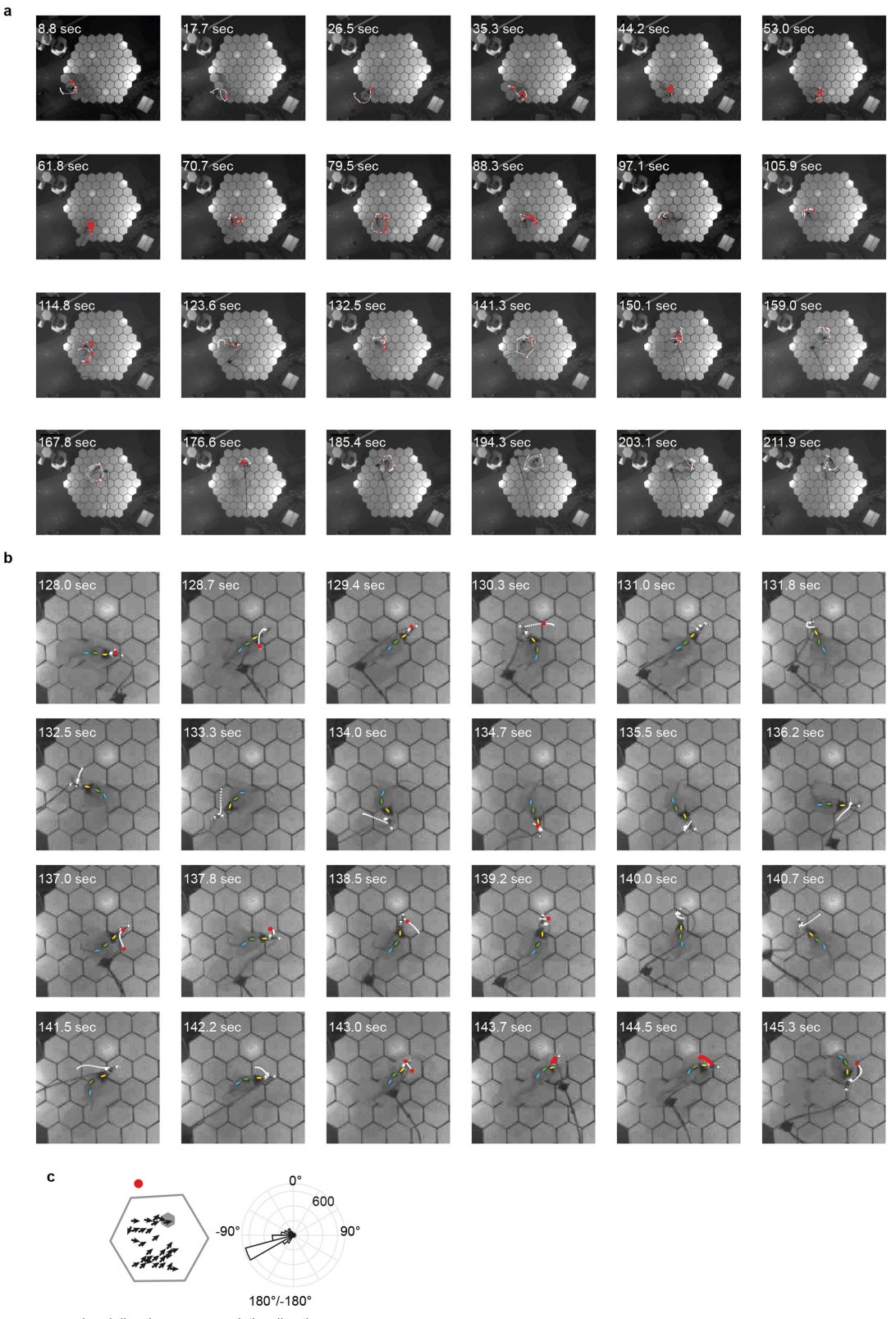

**Extended Data Fig. 2 | Typical behaviour and spiking of a ConSink neuron.**
**a**, Screengrabs of a complete trial of the honeycomb task (Rat 1, trial 3). In white
is the path of the animal from the time of the previous screengrab (or from 0 s
for the first screengrab) to the time of the current screengrab. In red are the
spikes fired by a representative ConSink cell (TT15c6). **b**, Screengrabs from a
portion of the same trial in (**a**) at a higher temporal resolution. Note how the
animal completes a full 360° rotation while waiting for the next pair of
platforms to be presented (platforms start to rise at 142sec); this is typical of all
5 animals' behaviour. 3 ellipses are drawn over the animal's cervical (yellow),
thoracic (green), and lumbar (blue) regions to show his position and
orientation. **c**, The same cell's vector field (left) and the polar plot of spike
directions relative to the convergence sink (right; the sink is plotted in red at
left).

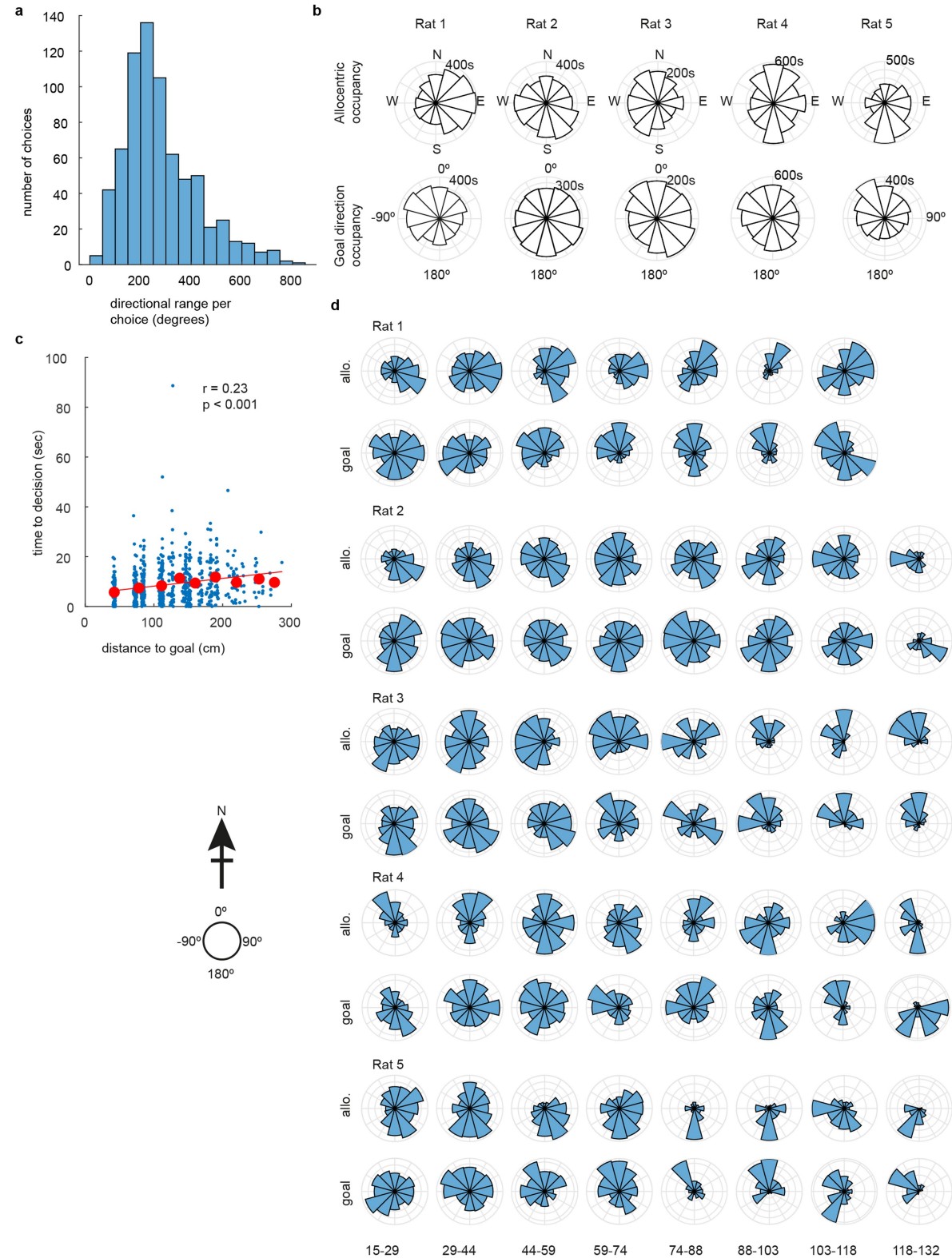

**Extended Data Fig. 3 | Rats sample a large range of possible directions while navigating the honeycomb task. a,** Histogram showing the directional range covered by the animals per choice from the time when the platforms begin to rise to the time when the animal moves to the next platform. Note that ranges greater than 360° indicate that the animal continued to scan in the same direction (i.e. multiple rotations); if an animal scanned 360° and then counter-rotated back to the starting direction, range is calculated only as 360°. **b,** Allocentric (allo.) and relative-to-goal (goal) directional occupancy for each animal. Note that goal direction is not oversampled. **c,** The time the animals take to make their decision decreases as they get closer to goal (one sample t test, t(723) = 6.27, p = 5.94 x 10⁻¹⁰); however, this does not seem to prevent them sampling the full range of direction at short-goal distances (**d**).

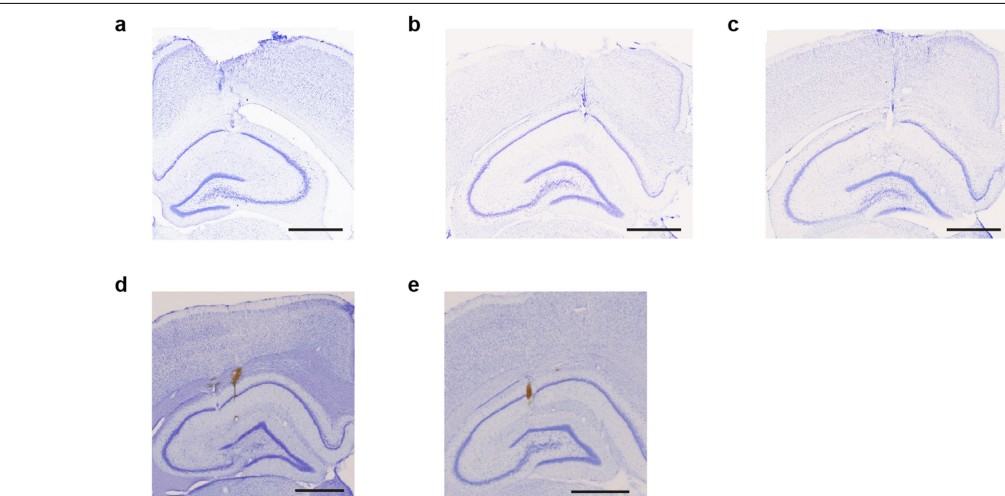

**Extended Data Fig. 4 | Histology showing tetrode position in the CA1 cell layer in dorsal hippocampus.** Example cresyl violet stained coronal slices showing tetrode marking lesions in all 5 experimental subjects (Rat 1 (**a**), 2 (**b**), 3 (**c**), 4 (**d**), and 5 (**e**)). Scale bar = 1000 μm.

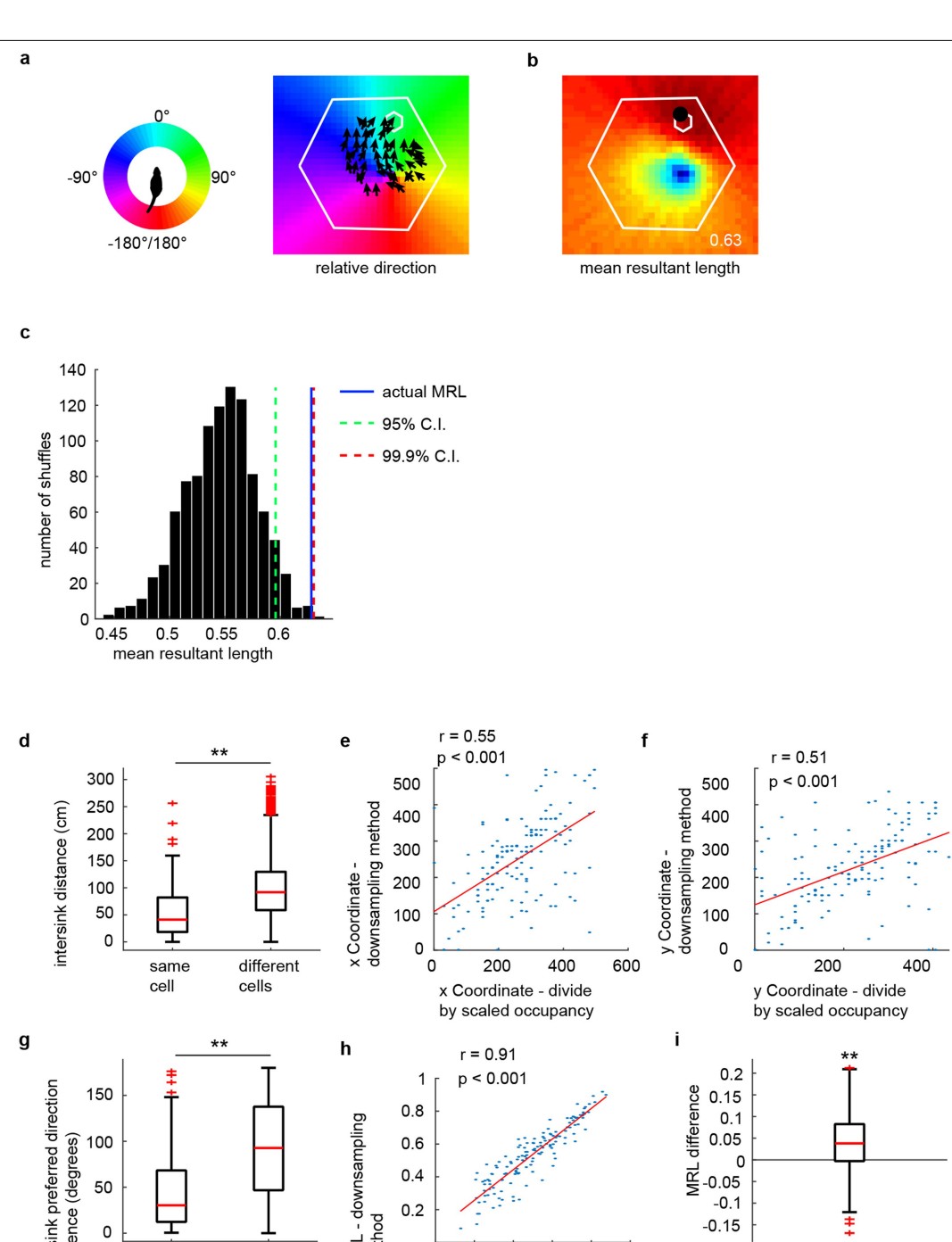

**Extended Data Fig. 5** | See next page for caption.

**Extended Data Fig. 5 | Calculation of the ConSink. a**, Left, each division of the colour wheel represents 1 search position in a polar co-ordinate framework centred on the animal's head with 0° straight ahead. Right, the spatial environment is tiled by candidate sink positions. At each candidate position, the direction of all spikes relative to that position can be calculated (by subtracting the direction of the vector from the animal to the candidate position from the animal's allocentric head direction). From this, a mean direction can be calculated, and is plotted here according to the colour wheel at left. The vector field (i.e. the mean allocentric direction of spikes at binned spatial positions) of an example ConSink cell (Rat 2, cell TT18c1) is overlaid. **b**, At each candidate sink position, the mean resultant length (MRL) of the associated distribution of relative directions is calculated (candidate positions are colour coded by MRL value). The candidate position with the largest MRL, which indicates the concentration of the polar distribution in the mean direction for that position, is taken as the ConSink (black closed circle; MRL = 0.63). **c**, To determine whether a cell was significantly modulated by relative direction to the candidate position identified as in (**b**), we shuffled the allocentric head directions associated with each spike, and recalculated relative direction MRLs at each search position, using the maximal MRL for our control distribution. This procedure was repeated 1000 times, and confidence intervals constructed. A cell was deemed to be significantly modulated if it's MRL was greater than the 95$^{th}$ percentile of the shuffled distribution of maximal MRLs. **d**, To confirm the validity of our calculations, we recalculated the sinks in our identified ConSink cells using a downsampling method, in which, for each cell on each platform, the spikes were downsampled according to the directional occupancy in allocentric coordinates (see Methods). We then compared the distances between the sinks calculated with the 2 different methods (our "divide by scaled occupancy" method, and "downsampling" method) within and between cells. We found that sink positions were more similar within cells than across cells (Wilcoxson rank sum test, two-sided, n = 142 cells (same cells) or 20,022 pairs of cells (different cells) from 5 animals, z = −10.04, p = 9.94 x 10$^{-24}$), and x (**e**, Pearson correlation, r = 0.55, p = 1.50 x 10$^{-12}$) and y coordinates (**f**, Pearson correlation, r = 0.51, p = 1.02 x 10$^{-12}$) were strongly correlated across the two techniques. **g**, Similarly, preferred relative direction was more similar within cells (Wilcoxson rank sum test, two-sided, n = 142 cells (same cells) or 20,022 pairs of cells (different cells) from 5 animals, z = −9.65, p = 5.07 x 10$^{-22}$). **h**, The strength of tuning was also highly correlated between the two techniques (Pearson correlation, r = 0.91, p = 7.99 x 10$^{-55}$). **i**, Lastly, we found that our technique did not overestimate the strength of tuning, as tuning was slightly, but significantly, stronger in the downsampled data (Wilcoxson signed rank test, two-sided, n = 142 cells from 5 animals, z = 5.60, p = 2.18 x 10$^{-8}$). For box plots in (**d**), (**g**) and (**i**), the central mark indicates the median, and the bottom and top edges of the box indicate the 25th and 75th percentiles, respectively; the whiskers extend to the most extreme data points within 1.5 times the interquartile range away from the bottom or top of the box, and all more extreme points are plotted individually using the '+' symbol.

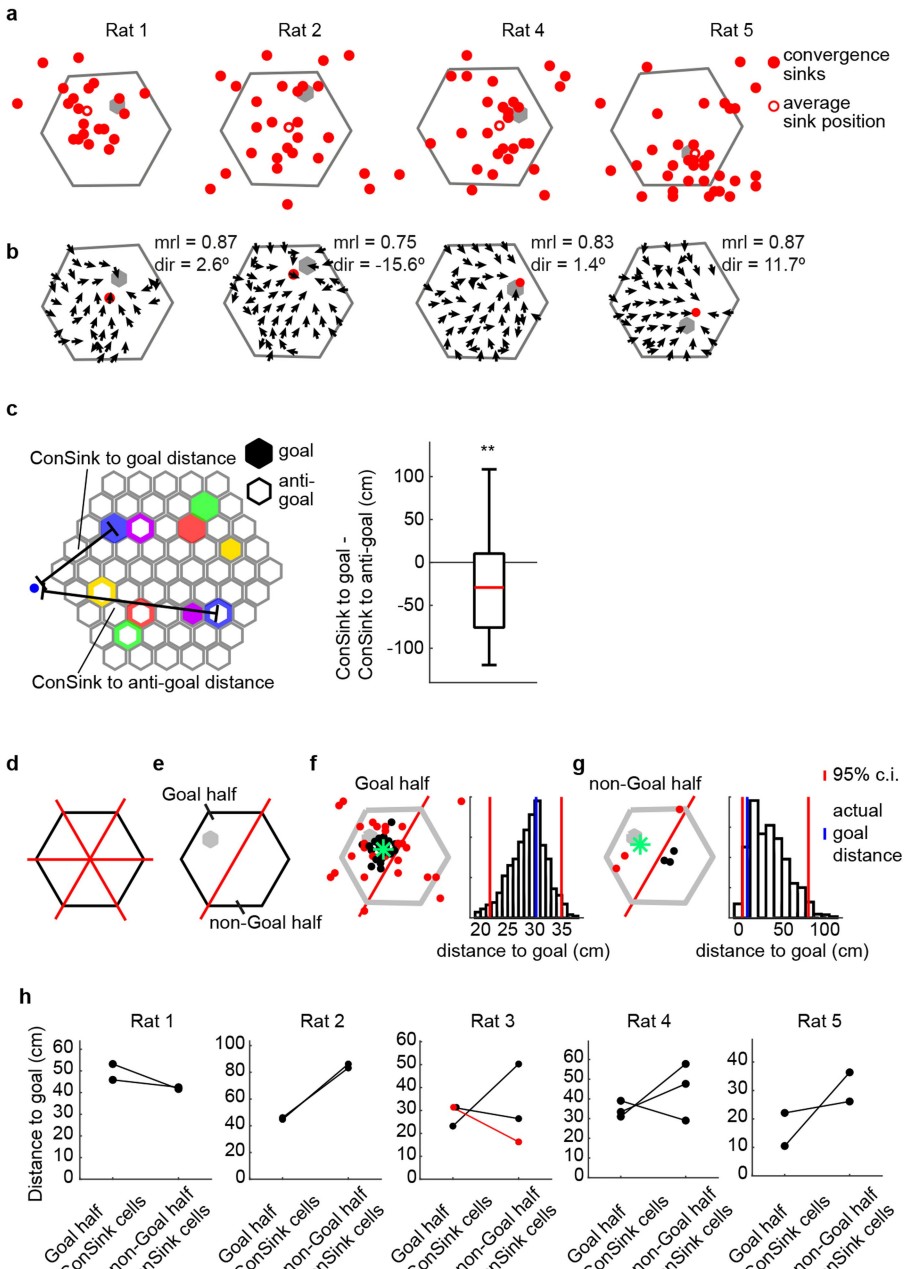

**Extended Data Fig. 6 | ConSinks cluster around the goal. a**, ConSinks from Rat 1, 2, 4 and 5 plotted over the maze showing that conSinks are widely distributed across the maze, and some are also located past the maze perimeter (see Fig. 1 for Rat 3). Grey hexagons represent goal platforms. **b**, Average vector fields for Rat 1, 2, 4, and 5 (see Fig. 1 for Rat 3). "mrl", mean resultant length; "dir", mean relative direction. **c**, Left, Schematic of the maze showing positions of goals (red: Rat 1; green: Rat 2; blue: Rat 3; yellow: Rat 4; purple: Rat 5) and platforms used as anti-goals in the analysis at Right. The anti-goal positions were produced by mirroring the goal positions across the axis perpendicular to a line between the maze vertex closest to the goal and the opposite vertex. For each relative direction cell, the distances from its convergence sink to the goal and anti-goal were calculated. The differences between these distances is plotted at Right. Convergence sinks were closer to the goal than the anti-goal, suggesting greater density around the goal (Wilcoxon signed rank test, two-sided, n = 142 cells from 5 animals, z = −4.92, p = 8.61 x 10⁻⁷. For box plot, the central mark indicates the median, and the bottom and top edges of the box indicate the 25th and 75th percentiles, respectively; the whiskers extend to the most extreme data points away from

the bottom and top of the box). **d**, To determine whether clustering around the goal was due to place field locations (taken as the centre of mass), the maze was divided into 2 halves along 3 different axes, producing 3 pairs of halves. **e**, The half containing the goal is referred to as the "Goal half", the other as the "non-Goal-half" (for Rats 1, 2, and 5, one pair of halves is eliminated from the analysis as their goals lie on 1 of the 3 axes). **f**, Left, the place cell positions of the Goal half cells corresponding to the split axis (red line) from Rat 3 shown as black filled circles. The ConSink positions of the same cells shown as red filled circles. The average ConSink position shown as a green asterisk. Goal in grey. Right, Histogram of average ConSink distances to goal calculated by randomly sampling the same number of cells as shown at left, but from the whole maze, repeated 1000 times. The red bars delimit the 95% confidence interval. The goal distance of the average ConSink at left is shown in blue. **g**, Same as **f**, but for the non-goal half. **h**, The distances to goal of each of the 2 or 3 pairs of Goal-half and non-Goal-half average ConSinks for all 5 rats. Note that the significance of the difference for each pair of points was calculated using the bootstrap method shown in (**d**) and (**e**); none fell outside the 95% confidence intervals. n = 2 (rats 1, 2 and 5), 3 (rats 3 and 4).

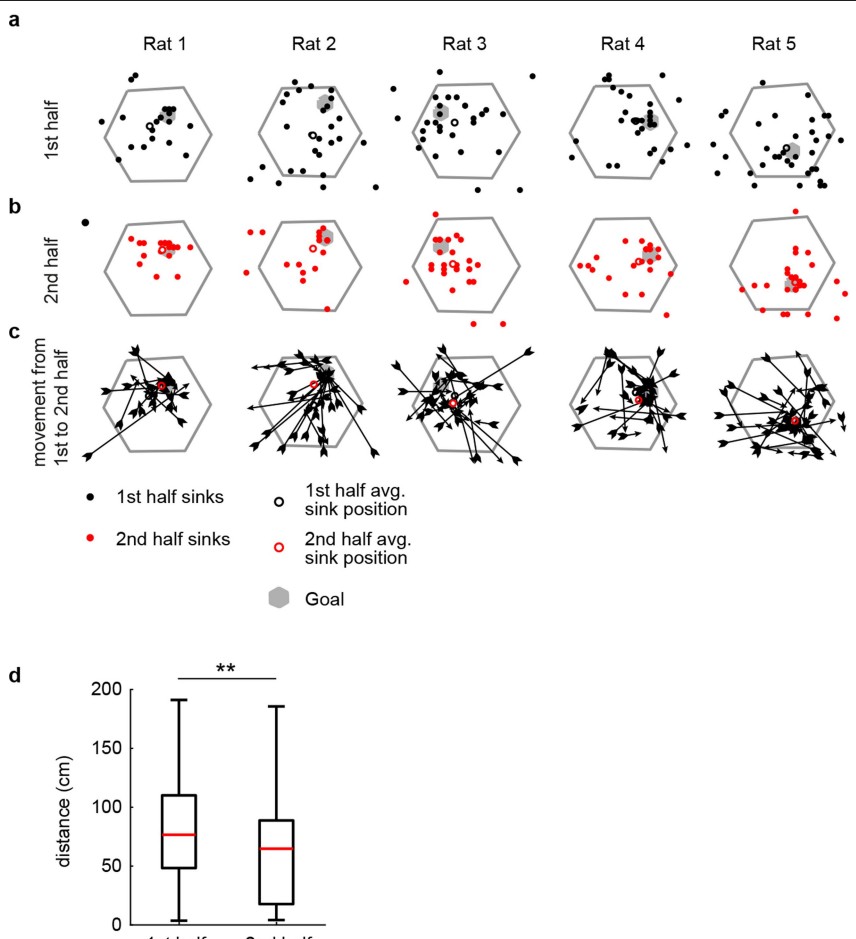

**Extended Data Fig. 7 | ConSinks move closer to the original goal with experience within a day.** Compared to ConSinks calculated during the first half of the first recording session (**a**), ConSinks calculated during the second half of the first session (**b**) appear more concentrated around the goal. **c**, Most ConSinks move towards the goal. Arrowheads refer to locations of ConSinks in (**b**), tails to ConSinks in (**a**). **d**, The second half ConSinks are significantly closer to the goal than first half ConSinks (Wilcoxon signed rank test, two-sided, n = 142 cells from 5 animals, z = 3.63, p = 2.86 x 10$^{-4}$. For box plot, the central mark indicates the median, and the bottom and top edges of the box indicate the 25th and 75th percentiles, respectively; the whiskers extend to the most extreme data points away from the bottom and top of the box).

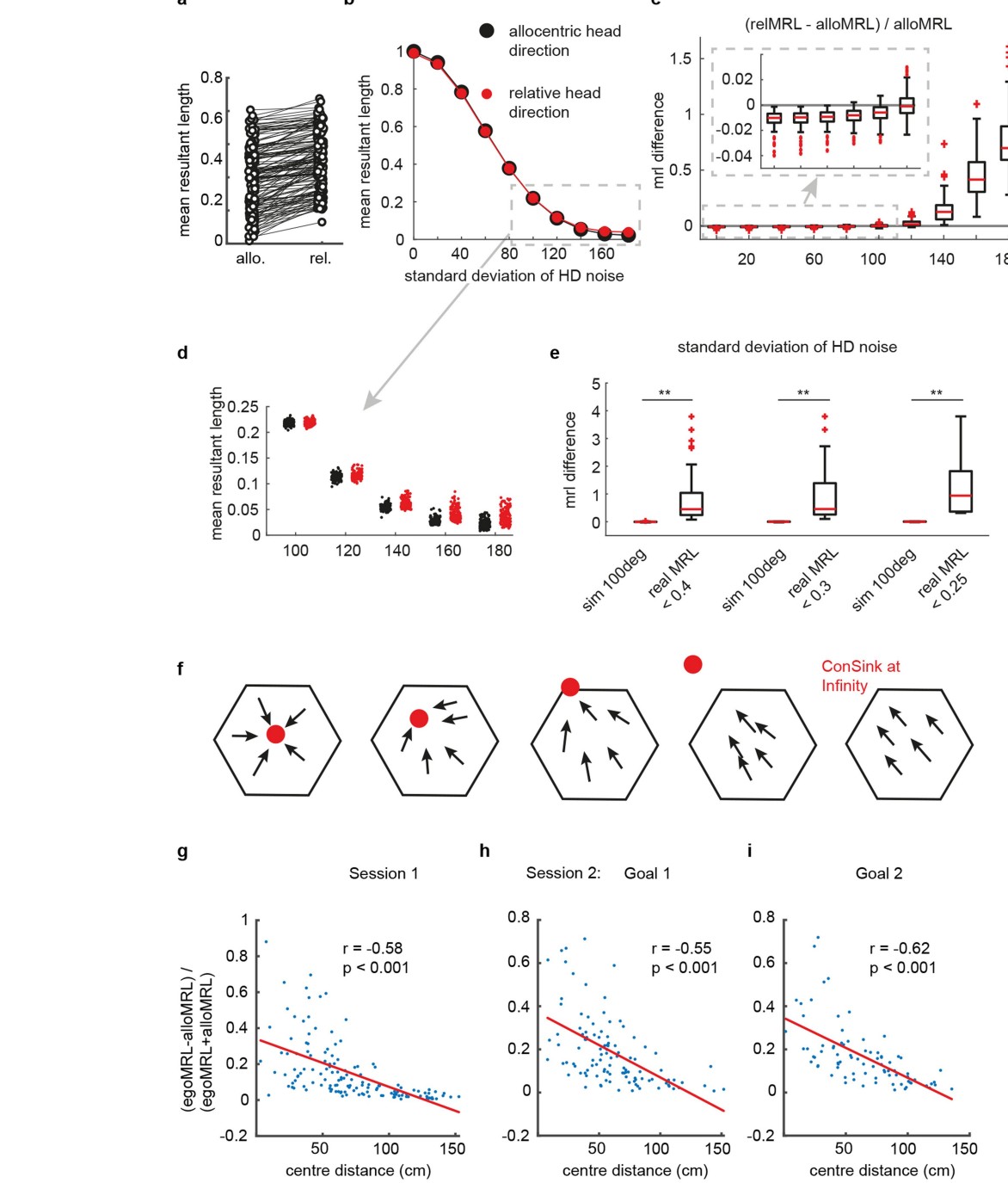

**Extended Data Fig. 8** | See next page for caption.

Extended Data Fig. 8 | Tuning to the ConSink is not an artefact of allocentric tuning in ConSink cells Part 1. a, MRLs calculated from allocentric (allo.) and relative (rel.) head directions for each ConSink cell during task. Note that for every ConSink cell, the MRL calculated from relative directions is greater than the MRL calculated from allocentric head directions. n = 142 cells from 4 animals. b-e, Noise in a purely allocentric head direction signal can't explain the greater egocentric MRLs in ConSink cells. b, We simulated purely allocentric cells by assigning new head direction values to the spikes fired by each ConSink cell. These head directions were calculated by using a given cell's true mean allocentric head direction and adding increasing levels of noise. We then calculated both allocentric and egocentric MRLs as was done for our real data. In these simulated allocentric cells, egocentric MRLs closely tracked allocentric MRLs, and both decreased with increasing levels of head direction noise. c, Only at noise levels above 100deg standard deviation do egocentric MRLs become larger than allocentric MRLs. n = 142 cells from 5 animals. d, The MRLs at 120deg noise and above were smaller than any we observed in our identified ConSink cells, and thus irrelevant. e, Only in cells with ~100deg of head direction noise, and therefore MRLs of ~0.2 length, would we expect true allocentric cells to have relatively larger egocentric MRLs within the range of values we observed in our data. However, the differences in egocentric MRLs relative to allocentric MRLs in our most weakly tuned ConSink cells were still much greater than could be explained by noise in a purely allocentric signal (Wilcoxon rank sum test, two-sided; left: n = 42 cells from 5 animals, z = −7.89 p = 3.12 x 10$^{-15}$; middle: n = 24 cells from 5 animals, z = −5.93, p = 3.06 x 10$^{-9}$; right: n = 9 cells from 5 animals, p = 4.11 x 10$^{-5}$). f, Schematic showing how egocentric tuning appears more allocentric with increased ConSink centre-distance; this is due to the animal's narrow sampling of allocentric direction when oriented in the cell's optimal egocentric direction. This leads to the prediction that if ConSink cells are truly tuned to egocentric direction, the difference between the calculated strength of their egocentric and allocentric tunings (see (a)) will decrease with distance from the centre of the maze. g-i, There was a strong negative correlation between egocentric-allocentric tuning difference and distance to the maze centre in both session 1 and session 2 (g, session 1, Pearson correlation, r = −0.58, p = 2.86 x 10$^{-14}$; h, session 2-goal 1, Pearson correlation, r = −0.55, p = 7.98 x 10$^{-10}$; i, session 2-goal 2, Pearson correlation, r = −0.62, p = 4.82 x 10$^{-10}$). For box plots in (c) and (e), the central mark indicates the median, and the bottom and top edges of the box indicate the 25th and 75th percentiles, respectively; the whiskers extend to the most extreme data points within 1.5 times the interquartile range away from the bottom or top of the box, and all more extreme points are plotted individually using the '+' symbol.

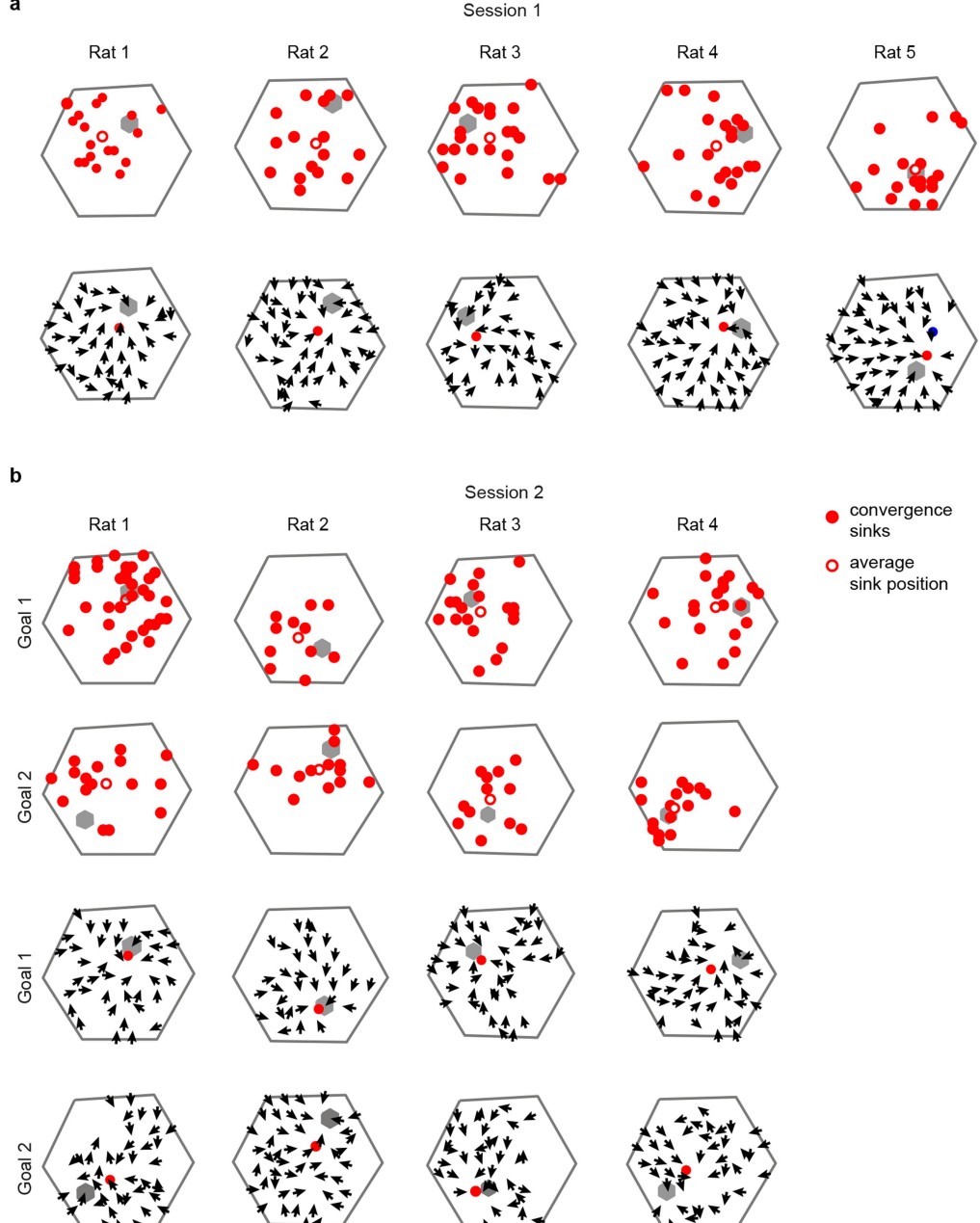

**Extended Data Fig. 9 | Tuning to the ConSink is not an artefact of allocentric tuning in ConSink cells Part 2. a, b,** Because it is more difficult to distinguish whether cells with sinks off the maze were egocentric or allocentric, we recalculated mean sink positions and population field vectors using only ConSink cells with sinks on the maze (**a**, Session 1, **b**, Session 2; number of cell omitted: rat 1, session 1: 3/21; session 2 goal 1: 8/47, goal 2: 9/25; rat 2, session 1: 10/27; session 2 goal 1: 3/14, goal 2: 8/21; rat 3, session 1: 7/29; session 2 goal 1: 4/23, goal 2: 4/16; rat 4, session 1: 8/30; session 2 goal 1: 4/25, goal 2: 3/19; rat 5, session 1: 17/35). Mean sinks continued to move with the goals and field vectors continued to point to the goals.

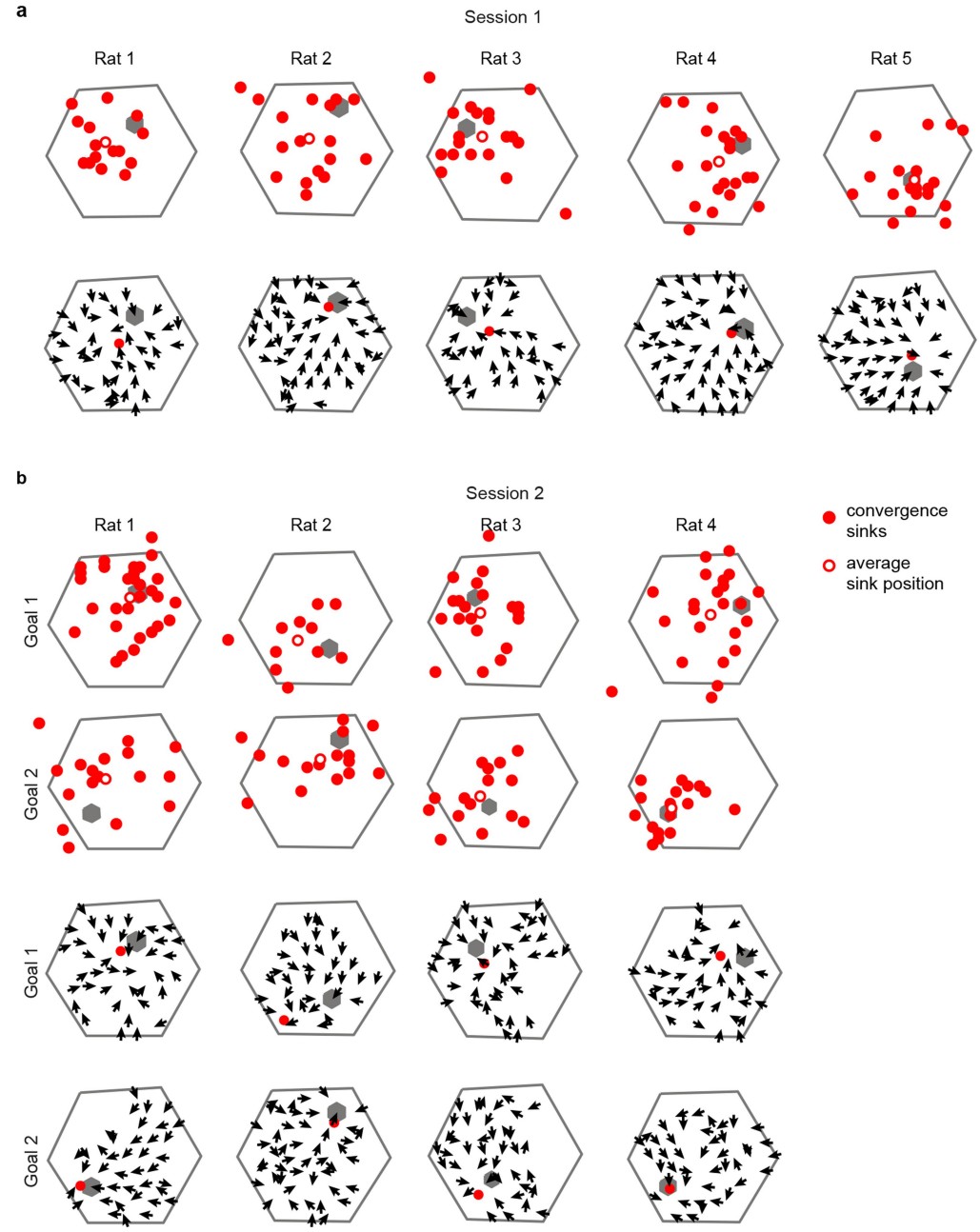

**Extended Data Fig. 10 | Tuning to the ConSink is not an artefact of allocentric tuning in ConSink cells Part 3. a, b,** We repeated this analysis omitting those cells with a normalized egocentric-allocentric MRL difference of 0.05, that is, those ConSink cells most likely to be mis-identified as allocentric-tuned cells (number of cells omitted: rat 1, session 1: 6/21; session 2 goal 1: 13/47, goal 2: 8/25; rat 2, session 1: 10/27; session 2 goal 1: 3/14, goal 2: 5/21; rat 3, session 1: 10/29; session 2 goal 1: 3/23, goal 2: 1/16; rat 4, session 1: 7/30; session 2 goal 1: 1/25, goal 2: 2/19; rat 5, session 1: 16/35). Similarly, there was little change in the mean sinks or the field vectors.

# Reporting Summary

## Statistics

For all statistical analyses, confirm that the following items are present in the figure legend, table legend, main text, or Methods section.

| n/a | Confirmed | |
|---|---|---|
| ☐ | ☒ | The exact sample size ($n$) for each experimental group/condition, given as a discrete number and unit of measurement |
| ☒ | ☐ | A statement on whether measurements were taken from distinct samples or whether the same sample was measured repeatedly |
| ☐ | ☒ | The statistical test(s) used AND whether they are one- or two-sided<br>*Only common tests should be described solely by name; describe more complex techniques in the Methods section.* |
| ☒ | ☐ | A description of all covariates tested |
| ☒ | ☐ | A description of any assumptions or corrections, such as tests of normality and adjustment for multiple comparisons |
| ☐ | ☒ | A full description of the statistical parameters including central tendency (e.g. means) or other basic estimates (e.g. regression coefficient) AND variation (e.g. standard deviation) or associated estimates of uncertainty (e.g. confidence intervals) |
| ☐ | ☒ | For null hypothesis testing, the test statistic (e.g. $F$, $t$, $r$) with confidence intervals, effect sizes, degrees of freedom and $P$ value noted<br>*Give P values as exact values whenever suitable.* |
| ☒ | ☐ | For Bayesian analysis, information on the choice of priors and Markov chain Monte Carlo settings |
| ☒ | ☐ | For hierarchical and complex designs, identification of the appropriate level for tests and full reporting of outcomes |
| ☐ | ☒ | Estimates of effect sizes (e.g. Cohen's $d$, Pearson's $r$), indicating how they were calculated |

*Our web collection on statistics for biologists contains articles on many of the points above.*

## Software and code

Policy information about availability of computer code

| Data collection | Custom code written in LabView 2018 was used to collect video data. Eletrophysiology data was collected using commercial software (Intan RHD data acquisition software Version 1.5.2). |
|---|---|
| Data analysis | Data analysis used custom code written in Matlab 2019a. Spike sorting performed using KiloSort (version 1) and Phy 2.0. Animal tracking performed in DeepLabCut2.0. |

For manuscripts utilizing custom algorithms or software that are central to the research but not yet described in published literature, software must be made available to editors and reviewers. We strongly encourage code deposition in a community repository (e.g. GitHub). See the Nature Portfolio guidelines for submitting code & software for further information.

## Data

Policy information about availability of data

All manuscripts must include a data availability statement. This statement should provide the following information, where applicable:

- Accession codes, unique identifiers, or web links for publicly available datasets
- A description of any restrictions on data availability
- For clinical datasets or third party data, please ensure that the statement adheres to our policy

The datasets generated during the current study are available from the corresponding authors on reasonable request. Source data are provided with this paper.

# Field-specific reporting

Please select the one below that is the best fit for your research. If you are not sure, read the appropriate sections before making your selection.

☒ Life sciences          ☐ Behavioural & social sciences          ☐ Ecological, evolutionary & environmental sciences

For a reference copy of the document with all sections, see nature.com/documents/nr-reporting-summary-flat.pdf

# Life sciences study design

All studies must disclose on these points even when the disclosure is negative.

| | |
|---|---|
| Sample size | No statistical methods were used to pre-determine sample size. Numbers of animals used were based on those used in the authors' previous publications as well as other studies in the same field (e.g. Ormond and McNaughton, PNAS, 2015; Wang, Foster, and Pfeiffer et al., Science, 2020). |
| Data exclusions | No data were excluded. |
| Replication | ConSink cells were identified in all 5 animals used in the study. The findings reported in Figures 1-3 (and associated Extended Data figures) were presented for each individual animal (in addition to grouped population analyses). |
| Randomization | Our study did not use separate groups, so randomization was not used. |
| Blinding | Our study did not use separate groups, so blinding was not used. |

# Reporting for specific materials, systems and methods

We require information from authors about some types of materials, experimental systems and methods used in many studies. Here, indicate whether each material, system or method listed is relevant to your study. If you are not sure if a list item applies to your research, read the appropriate section before selecting a response.

### Materials & experimental systems

| n/a | Involved in the study |
|---|---|
| ☒ | ☐ Antibodies |
| ☒ | ☐ Eukaryotic cell lines |
| ☒ | ☐ Palaeontology and archaeology |
| ☐ | ☒ Animals and other organisms |
| ☒ | ☐ Human research participants |
| ☒ | ☐ Clinical data |
| ☒ | ☐ Dual use research of concern |

### Methods

| n/a | Involved in the study |
|---|---|
| ☒ | ☐ ChIP-seq |
| ☒ | ☐ Flow cytometry |
| ☒ | ☐ MRI-based neuroimaging |

## Animals and other organisms

Policy information about studies involving animals; ARRIVE guidelines recommended for reporting animal research

| | |
|---|---|
| Laboratory animals | 5 male Lister Hooded rats, aged 9-12 months, purchased from Charles River Laboratories. |
| Wild animals | n/a |
| Field-collected samples | n/a |
| Ethics oversight | Experiments were carried out in accordance with British Home Office Regulations, and study protocols were reviewed by the Animal Welfare and Ethical Review Board at University College London. |

Note that full information on the approval of the study protocol must also be provided in the manuscript.

