## [Peer Review File · Nature]

Manuscript Title: Hippocampal place cells exhibit goal-oriented vector fields during navigation

Reviewer Comments & Author Rebuttals

Reviewer Reports on the Initial Version:

Referees' comments:

Referee #1 (Remarks to the Author):

This article presents the original and significant concept of convergence sinks that can describe variations in the firing rate of hippocampal place cells during performance of a goal-directed navigation task. The convergence sink provides a description of variation in the firing rate of place cells based on the relative angle and distance of the animal's current heading and location and the hypothesized convergence sink. This provides an exciting additional perspective on the functional role of place cells by adding a vector representation that can be used to guide navigation to a goal. The convergence sinks appear to be influenced by the current goal location, as the sum of the convergence sinks over place cells is close to the first goal location (Fig. 2), and when the first goal location is replaced by a second goal location, the sum of convergence sinks shifts to a new location (Fig. 3), and over time of training on the second goal location, the sum of convergence sinks moves even closer to the second goal location. In addition, the average vector fields pointing toward the convergence sink for significant ConSink cells seems to describe a vector field directional fantail leading to the goal location, which could be used to guide choices (Fig. 4) and which shows less accurate heading toward the goal on error choices (Fig. 5). These are novel and important results suggesting mechanisms for the role of the hippocampus in goal-directed spatial navigation.

Major comment:

1. This new perspective on the modulation of place cell firing rate for goal directed navigation is very interesting and important. However, it is somewhat surprising that the convergence sinks are so broadly distributed in the environment. In addition, it seems somewhat surprising that the experimental recording of a relatively small number of neurons with convergence sinks, and such wide distribution of convergence sinks locations will show an average close to the goal location. It would be interesting to see if this depends upon the distribution of the place cell locations themselves. For example, if an even smaller number of place cells are selected that cover a broad range of locations, will these converge on the goal location, whereas selection of place cells with field locations that are biased toward one half of the environment would not?

2. The relationship between Figure 1h and Figure 2a is unclear, as initially it seems that both are showing goal location (gray hexagon) and the location of the average ConSink (black dot in Fig. 1h and open circle in Fig. 2a), but then it is not clear why these differ between the two figures. The text says "a goal shift experiment on a subsequent day," but my initial assumption was that the familiar location (goal 1) in the first 13 trials of that day (Fig. 2a) would be the same goal location used in previous studies (Fig. 1h). The methods state that "all animals ran 13 trials to goal 1," which again

seems to be referring to the goal 1 from previous days and does not contradict this initial assumption (see specific comments below). It would help to have a legend or text description of what the gray hexagons represent in the two figures (I presume goal location, but it is not stated), and to have the main text and methods explicitly clarify that the goal was moved between the goal location in Fig. 1 and the “goal 1” in Fig. 2.

Specific comments:

Line 7 - Somehow it seems odd to capitalize Hippocampal Cognitive Map, though it is clearly an important concept in this field.

Line 22 - “ConSinks” is not explicitly defined – should appear earlier with “convergence sinks” on line 20.

Line 58 – “of these 77 (29%)” – In addition to giving the sum, they should present the number of ConSink cells recorded in each individual rat as the calculation of mean convergence sink (Fig. 1h) and relation to goal location depends upon averaging within each rat.

Line 110 – “no such shift was seen” – should specify that the shift mentioned in the preceding sentence was specifically for ConSink cells.

Line 251 – “the centre of the environment” – the egocentric coding toward centre of the environment was also shown earlier by Wang et al. 2018 from the Knierim lab, but they might mention that this measure can be ambiguous relative to coding of environmental boundaries as shown in recordings of retrosplenial cortex by Alexander et al., 2020.

Line 346 – Is there a difference in calculation technique between the average sink position (black dot) in Fig. 1h and the average ConSinks shown with open circles in Fig. 2a? Perhaps the difference is only due to this being different experiments. But Figure 1h itself labels the black dot as “average sink position” whereas the figure legend says “centroid” so perhaps this indicates a difference.

Line 348 – Fig. 1i “average vector fields” – are these the vectors for individual ConSink place cells or the average across place cells for a location? It is confusing whether or not this is based on all cells firing in a location because the vector fields are shown on a cell by cell basis for significant ConSink cells in Figs. 2g and 2h.

Line 392 – What is the thin black line in Figure 4L? Is it the average (fantail) of the three rats? If not, could they plot the fantail here directly for comparison?

Line 466 – “ In the second recording session.” – This phrase appears twice .

Line 467 – “To goal 1” – this reads as if it is the same goal used in the first recording session (which was the assumption I made based on the main text). If not, it should be emphasized here AND in the main text that goal 1 on the shift goal task is NOT the same as the initial goal used in Figure 1. Maybe it would be better not to call it goal 1 as that suggests it is the first goal in the overall experiment, whereas it might just be the first goal in the two goal task?

Line 576 – “all animals ran 13 trials to goal 1” – this is still confusing as it suggests goal 1 is the goal in Fig. 1, but the goal locations don’t seem to match between Fig. 1 and Fig. 2.

Line 715 – it is hard to see the rat in Extended Data Fig. 2. Is there a chance that an outline could be drawn or a line could be drawn down the midline of the rat to ease viewing. It is also very hard to tell that it is on a raised platform, which is surprising as it suggests the camera aligns the borders of the raised platform perfectly with the lower platforms.

Referee #2 (Remarks to the Author):

This paper investigates an important question in the study of navigation, addressing the mechanism and computations used by the hippocampus to support goal-directed navigation. Using their newly developed honeycomb maze, the authors demonstrate that CA1 place cells display egocentric vector coding between the animal’s current location and the goal location. The authors further show that, in principle, all of the information that the animal needs to guide its behavior to the goal at every step is contained in this vector code.

Although the demonstration of egocentric coding in place cells and other cells in the medial temporal lobe system is not novel, the authors relate this coding to behavior in an interesting way. Their conclusions about vector coding are important. However, there are a number of concerns with the data analyses that weaken the case, to the point that it is not clear whether their conclusions are supported strongly enough. These concerns would have to be addressed in order to make the case compelling enough for publication.

Major

1) p. 3, “Of these, 77 (29%) displayed firing patterns within the place field which were best described by vector fields converging on a location that, following vector field notation, we term a convergence sink.” I assume that this is a comparison with allocentric head direction tuning. However, the authors comparison between the egocentric coding and allocentric HD coding (Extended Data Fig. 5b) appears biased to favor egocentric coding. For each cell, multiple egocentric bearing tuning curves are calculated, one for each of the 986 (34 x 29) potential ConSinks. The egocentric curve with the best ConSink is chosen. However, there is only one allocentric HD tuning curve (by definition). Assuming that there is some amount of noise and stochasticity in the firing, a place cell with allocentric HD tuning (e.g., perhaps Cell g of Figure 1) would show a well-tuned curve with any of the more peripheral ConSinks, and the algorithm would pick the best one out of many (ie, the extreme tail of the distribution). Because of this biased selection, chances are high that the best egocentric tuning curve will be tighter than the single allocentric tuning curve. The authors need to make a fair comparison between the ConSink-based tuning curves and the allocentric tuning curve to state with certainty how many of the apparent ConSink cells are actually allocentric HD cells (or cannot be statistically distinguished from HD cells). It is not readily apparent to this reviewer how to do this, but it is a critical component.

2) I would suggest that the authors do away with the terminology of a “ConSink cell.” There is a

proliferation of names that identify particular properties of cells with a classification of a new cell type. Although this makes sense when describing clearly different types of cells (eg., place cells, grid cells, head direction cells, border cells, etc.), it adds confusion when a newly discovered property of these cell types is called a new cell type. Furthermore, in these other cases, the term reflects the putative representational function of the cell (e.g., a place cell represents a place in the environment, a grid cell represents a grid-like coordinate frame, etc.). What does a ConSink cell represent? The ConSink? I don't believe that the authors think this. (Similar concerns apply to other authors' usage of the term "anchor cell" in Reference 12 to refer to a very similar property as that describe in the present paper.)

3) P. 4: "Similarly, the vector fields and their associated population ConSinks moved from the original goal locations towards the new goals." This is clear only for Rat 3. Rats 1 and 2 do not show an obvious shift toward the goal. The apparent shift could be a statistical artifact; that is, given that the average ConSink is initially near Goal 1, assume that the average shifts to a new, random location for Goal 2. Because $\sim 2/3$ of the platform encompasses locations closer to the new goal rather than farther away from it (relative to the initial position), the analysis is biased in favor of this effect. The authors need to estimate the probability of the vector field average goal shifting closer to Goal 2 by chance and use this as the null hypothesis.

4) P. 10: The authors state in the Discussion that "the animal simply needs to move in the direction of highest firing rate afforded by the choices available, essentially comparing the lengths of the available branches of the Fantail." How is this done computationally based on these results? Does the animal do a VTE behavior? Can the authors see these differences in population firing rate? The conclusions of this paper would be much more compelling if the authors could show evidence of this comparison taking place while the animal was deliberating its choice. Without this analysis, the conclusions are much weaker and uncertain.

5) Figure 1i: How are the average vector fields calculated? Summing all spikes from a recording and dividing by time? Calculating a vector for each cell and location and calculating the mean vector? Or some other algorithm? (Apologies if this is in the Methods but I missed it.)

6) The finding that the mean direction of significant ConSinks was indistinguishable from 0 suggests that these cells are really just place cells that fire most strongly when the rat is pointing toward the goal location, not a more general vector coding scheme. Related findings are found in Acharya et al. Cell (2016) from Mehta's group, in which directional firing toward cue locations was revealed in their VR system that disrupts the spatial firing of place cells. The same inputs to pyramidal cells that produces this type of cue-directed directional tuning may also produce goal-directed firing in place cells in the present task.

7) p. 29: A shuffling procedure is not technically a bootstrap. A bootstrap is a specific procedure in which an underlying population distribution is estimated by multiple iterations of random sampling with replacement of the same number of observations as in the data sample. This is not what the authors performed here. In addition, please provide more details of the exact shuffling methods used. These details are necessary for the read to evaluate the validity of the shuffling procedure and whether it is a reasonable representation of the null distribution based in the experimental

hypothesis. Were the head directions assigned randomly? If so, how does this change the temporal statistics related to the animal's head direction behavior and does this affect the validity of the shuffling procedure? Or was the train of head direction measurements shifted relative to the train of spikes, or some other method?

Minor

8) p. 1: "The distribution of these convergence sinks was centred near the goal location,..." This is an overstatement, as it is convincingly true in only 1 of 3 rats.

9) p. 2: (but see 8)" The authors should also cite Rubin et al. J Neurosci 2014 from the Ulanovsky lab

10) p. 2: ref 10—The authors should also cite Kobayashi et al J Neurophysiol 1997 from Ono lab.

11) p. 2: "human" The Kunz paper is not from dorsal hippocampus (or at least not mainly from the corresponding region in humans, ie, posterior hippocampus).

12) p. 4: "(282 CA1 place cells; Rat 1: 93 cells; Rat 2: 96 cells; Rat 3: 93 cells)" This seems very strange to have almost the identical number of cells for each rat. Is there something about the automated sorting method that sets a number of cells to identify, regardless of the isolation quality?

13) p/ 4: "32% (83/256) of principal cells during the original" Do the authors mean "place cells" rather than principal cells? For Fig 1 they said that 29% of "place cells" had ConSink tuning

14) p. 6: Extended Data Fig. 10c: The relative direction appears to represent 180 degrees less than the other directions. Is this significant?

15) p. 10: "Finally, we find that firing towards the goal or the individual sinks is reduced and the fantail pattern altered on error trials, indicating that ConSink cells are crucially involved in navigation the goal." This is an overstatement that implies causality, which has not been demonstrated in this paper. Although the data are highly suggestive, they do not demonstrate that these cells are "crucially involved."

16) Figure 2a, 2c: There appear to be fewer dots than indicated by the total number of cells mentioned in the text.

17) P. 24: Maze section. Please describe how the animal's behavior was shaped initially.

18) P 28-29: last 2 lines of p 28 and first 3 lines of p. 29 are not clear. What does it mean that "the distributions were summed according to the number of spikes the cell fired on each platform"? The next sentence also needs clarification as to how this analysis takes into account uneven sampling.

19) p. 29, last 3 lines: Was this procedure also performed for the allocentric HD cell plots for comparison?

20) Extended Data Fig 7e (caption): "indicating stability of sink position within this sub-population

across the goal switch.” I wouldn’t call this a good test of stability per se—it just indicates that the sink positions are not random.

Referee #3 (Remarks to the Author):

Ormond and O’Keefe record from dorsal hippocampal place cells while rats perform a goal-directed task on a maze in which navigation is constrained to stepwise decisions to move in one of two candidate directions. They report that, in addition to coding allocentric position in the environment, place cell activity reflects egocentric bearing to vector field convergence sinks (here dubbed ConSinks) which, at least when averaged together, are located near goal locations. Many such ConSinks were located far from the goal. Relative bearing tuning for goal locations was reduced during free foraging and on incorrect trials. The authors argue that the dual computation of egocentric orientation to a goal, allocentric orientation (relative to the recording room environment) and spatial position enables the hippocampus alone to solve the task. That is, the offset between the egocentric and allocentric orientations at any given location could be used to compute an optimal reorientation of the animal in the direction of the goal. The actual computation would, presumably, be carried out by a structure receiving CA1 output. Notably, one would have to assume that such a structure would determine the optimal reorientation from a large number of different CA1 ensemble patterns as these differ most robustly by environmental location. A number of related findings regarding goal-related directional modulation of place cell firing (Markus et al., 1995; Acharya et al., 2016; Sarel et al., 2017; Gauthier et al., 2018) have already been reported both in the hippocampus and parahippocampal areas (Wang, Chen, et al., 2018). Because of this, the true novelty of the work lies in providing the experimental context of an actual navigational behavior and a full-fledged consideration of a mechanism for it (combining location, distance to goal, egocentric orientation to goal, and allocentric orientation). Below are specific comments/concerns.

There should be considerably more analysis regarding the behavior of the animal on the task and how it relates to the convergence sinks observed. For instance, the authors claim at multiple points that the animal frequently samples all head directions while waiting on each platform but there is no explicit analysis of this. How does the animal’s time to choice, egocentric bearing, allocentric heading, etc. vary as a function of distance to the goal? How does the egocentric bearing to the goal or overall directional heading vary over learning of the goal location and how does it change when a new goal is introduced?

The full dataset (266 cells with significant spatial information scores) seems rather small as is the number of neurons exhibiting a ConSink (77). This issue with the work is exacerbated by the fact that the test for directional tuning has a rather low threshold for significance (resultant length vs that for a distribution of resultants following randomization of orientations relative to spikes). Similarly, the dataset is constrained with respect to the total number of trials (13-26) in any given recording.

The distribution of ConSinks in all 3 rats may, when averaged, reflect tuning to the goal locations. However, their spatial distributions are so broad that it seems implausible that an ensemble of CA1

neurons provides more than a 'lean' to the relative direction and distance to the goal. To be convincing as a mechanism actually guiding navigational behavior, one would hope for a far less noisy signal.

If a neuron has a punctate spatial receptive field (i.e. is a place cell) and the directional sampling is constrained (e.g. by more often facing the goal location) then one could imagine a scenario in which ConSinks emerge independent of actual directional sensitivity. This may relate to the overrepresentation of relative bearings directly in front of the animal. For this reason, the authors should consider downsampling to match head directions on each platform and rerunning a number of their analyses. Use of the Hardcastle model alone may not account for oversampling of specific headings which could cause considerable covariance of predictors related to egocentric goal bearing and allocentric heading or spatial position. Further, the GLM analysis relating to Figure 4 should be run on all cells, not just those possessing ConSinks. If place cells without ConSinks have significant sensitivity to the relative direction, allocentric direction, or distance to the goal what would that indicate about the presence of ConSinks and the subsequent conclusions? What happens if you run this model with bearing and distance calculated to any non-goal zone on the arena? If you take the small range of spatial positions represented by a place field and constrain the animal's head direction because of task structure you might produce egocentric coding to *any* position in the arena.

Claims regarding density of ConSinks near the goal and their movement relating to changes to goal position are not well supported by the data presented. The study should include an analysis to verify that the reported distance metrics and their relative changes are statistically different than a random distribution of pairwise comparisons between all other platforms.

The study should include a substantially expanded analysis of reliability. How much do ConSink positions change from trial to trial when there is no goal or when the goal is stationary for an extended period of time.

The fantail population analysis is underdeveloped and should be described in greater detail. Like other parts of this work one wonders if the distributions could be explained by behavioral biases of the animal. How much do the neural fantails resemble relative heading occupation?

Raw data points for all boxplots please

L122 Fig. 1j instead of i?

Spatial information, by itself, is a poor metric for identifying place cells in that it simply reflects bimodalities in the firing rates across locations. A measure, such as coherence, that reflects the clustering of high and low firing rates should be used to complement the spatial information measure. One can imagine simply using a threshold for number of spikes emitted to pare down the dataset. Why constrain the analyses to location-specific firing when two measures of orientation and distance are under consideration. Based on the analysis design, the work takes as an assumption the primacy of location in triggering firing of all CA1 neurons and should perhaps not.

With respect to the inclusion/exclusion of spikes occurring during sharp wave ripple events, the criteria seem reasonable, but the authors should compare the tuning for both kinds of spikes. Why presume that the relevant moments are all associated with theta patterned activity? It seems at least plausible that ripple-related activity could be more meaningful given the results of so many Foster laboratory publications.

The authors should generate a quantification (perhaps using coherence or diffusion metrics) that speaks to the concentration of high values in the maps of convergence resultant lengths. If ConSinks are to be taken very seriously, one would expect to see that a mapping convergence resultant lengths would look rather like the ratemap of a place cell.

Author Rebuttals to Initial Comments:

Reply to Referees' comments:

We thank the referees for their insightful and helpful comments and have answered all of them point by point and have incorporated most of the new materials into the paper.

Referee #1 (Remarks to the Author):

This article presents the original and significant concept of convergence sinks that can describe variations in the firing rate of hippocampal place cells during performance of a goal-directed navigation task. The convergence sink provides a description of variation in the firing rate of place cells based on the relative angle and distance of the animal's current heading and location and the hypothesized convergence sink. This provides an exciting additional perspective on the functional role of place cells by adding a vector representation that can be used to guide navigation to a goal. The convergence sinks appear to be influenced by the current goal location, as the sum of the convergence sinks over place cells is close to the first goal location (Fig. 2), and when the first goal location is replaced by a second goal location, the sum of convergence sinks shifts to a new location (Fig. 3), and over time of training on the second goal location, the sum of convergence sinks moves even closer to the second goal location. In addition, the average vector fields pointing toward the convergence sink for significant ConSink cells seems to describe a vector field directional fantail leading to the goal location, which could be used to guide choices (Fig. 4) and which shows less accurate heading toward the goal on error choices (Fig. 5). These are novel and important results suggesting mechanisms for the role of the hippocampus in goal-directed spatial navigation.

Major comment:

1. This new perspective on the modulation of place cell firing rate for goal directed navigation is very interesting and important.

However, it is somewhat surprising that the convergence sinks are so broadly distributed in the environment. In addition, it seems somewhat surprising that the experimental recording of a relatively small number of neurons with convergence sinks, and such wide distribution of convergence sinks locations will show an average close to the goal location. It would be interesting to see if this depends upon the distribution of the place cell locations themselves. For example, if an even smaller number of place cells are selected that cover a broad range of locations, will these converge on the goal location, whereas selection of place cells with field locations that are biased toward one half of the environment would not?

Response: As suggested by Referee 1, we divided the maze into halves, and assigned each ConSink cell to a half depending on the location of its place field centre of mass. Because the maze could be divided along different axes, we repeated the analysis 3 times along axes running from each corner to its opposite corner (panel b). The half containing the goal is referred to as the "Goal-half", and the other as the "non-Goal-half" (panel c, 3 halves of each type). For each half, we measured the distance from the average ConSink to the goal (panels d and e, left, black dots are PF centres, red dots are ConSink locations, red star is average ConSink location). We then conducted the control suggested by the reviewer,

randomly sampling the same number of cells as identified by place field centre location; this was repeated 1000 times to build a control distribution with confidence intervals (panels d, e right, red lines are 95% CI, blue line is distance from test group mean ConSink). We did not find any halves, either Goal or non-Goal, whose average ConSink was closer or further from the goal than expected by chance. Further, there wasn't a clear relationship between the goal distances of the Goal-half average ConSinks and their corresponding non-Goal half average ConSinks; in Rat 2, Goal-half average sinks were closer to the goal, consistent with the hypothesis proposed by Referee 1, but Rat 1 showed the opposite pattern, and Rat 3 varied depending on the splitting axis used (panel f). Thus, we do not find evidence of a relationship between place field position and average ConSink position. They appear to be independent attributes of the place cells.

We have added the figure to the Extended Data 6, and added the highlighted text (line 57):

“While ConSinks were scattered around the environment both on and off the maze, they were densest around the goal (Fig. 1h, Extended Data Fig. 6a); this clustering was not due to the positions of their place fields (Extended Data Fig. 6b-f).”

The figure legend now reads (line 794): “**b**, To determine whether clustering around the goal was due to place field locations (taken as the centre of mass), the maze was divided into 2 halves along 3 different axes, producing 3 pairs of halves. **c**, The half containing the goal is referred to as the “Goal half”, the other as the “non-Goal-half”. **d**, Left, The place cell positions of the Goal half cells corresponding to the split axis (red line) from Rat 3 shown as black filled circles. The ConSink positions of the same cells shown as red filled circles. The average ConSink position shown as a green star. Goal in grey. Right, Histogram of average ConSink distances to goal calculated by randomly sampling the same number of cells as shown at left, but from the whole maze, repeated 1000 times. The red bars delimit the 95% confidence interval. The goal distance of the average ConSink at left is shown in blue. **e**, Same as **d**, but for the non-goal half. **f**, The distances to goal of each of the 3 pairs of Goal-half and non-Goal-half average ConSinks for all 3 rats. Note that the significance of the difference for each pair of points was calculated using bootstrap method shown in (**d**) and (**e**); none fell outside the 95% confidence intervals.”

2. The relationship between Figure 1h and Figure 2a is unclear, as initially it seems that both are showing goal location (gray hexagon) and the location of the average ConSink (black dot in Fig. 1h and open circle in Fig. 2a), but then it is not clear why these differ between the two figures. The text says “a goal shift experiment on a subsequent day,” but my initial assumption was that the familiar location (goal 1) in the first 13 trials of that day (Fig. 2a) would be the same goal location used in previous studies (Fig. 1h). The methods state that “all animals ran 13 trials to goal 1,” which again seems to be referring to the goal 1 from previous days and does not contradict this initial assumption (see specific comments below). It would help to have a legend or text description of what the gray hexagons represent in the two figures (I presume goal location, but it is not stated), and to have the main text and methods explicitly clarify that the goal was moved between the goal location in Fig. 1 and the “goal 1” in Fig. 2.

Response: The referee is correct that it is a reasonable assumption that the goals used in the first session, presented in Figure 1, should be the same as Goal 1 from the goal-switch session presented in Figure 2. And we apologize for not making it clearer that this was indeed the case for rats 1 and 3 but not for rat 2. As stated in the methods, Rat 2 failed to learn the goal switch in our first attempt to carry out this procedure. Therefore, we had to retrain him in additional sessions to a new goal location, which is presented as Goal 1 in Fig. 2, before attempting a new goal switch experiment, which was successful and constitutes the data presented in Figure 2.

This is explained in the Methods. To clarify, we have now added the highlighted text at line 75:

“13 trials with a familiar goal (for Rats 1 and 3, the same goal as in Fig. 1, but for Rat 2, a new goal learned prior to session 2, see Methods) were followed, after some intermediary training, by another 13 trials to a new goal.”.

The reviewer is also correct that the grey hexagons represent the goals, and text explaining this has been added to the legends in Fig. 1h (line 340):

“Grey hexagons represent goal platforms.”

Fig. 2a is left unchanged as it already states that the goal is in grey.

Specific comments:

3. Line 7 - Somehow it seems odd to capitalize Hippocampal Cognitive Map, though it is clearly an important concept in this field.

Response: This has been corrected (line 7).

4. Line 22 - “ConSinks” is not explicitly defined – should appear earlier with “convergence sinks” on line 20.

Response: This has been corrected (line 16).

5. Line 58 – “of these 77 (29%)” – In addition to giving the sum, they should present the number of ConSink cells recorded in each individual rat as the calculation of mean convergence sink (Fig. 1h) and relation to goal location depends upon averaging within each rat.

Response: The text has been altered (line 50):

“Of these, 77 (29%; $n = 21$ (Rat 1), 27 (Rat 2), 29 (Rat 3)) displayed firing patterns...”

6. Line 110 – “no such shift was seen” – should specify that the shift mentioned in the preceding sentence was specifically for ConSink cells.

Response: The preceding sentence was altered to address this comment (line 94). It now reads:

“Place field centres of ConSink cells, on the other hand, while they did cluster around the original goal^{10,17,18} (Extended Data Fig. 14a-c), did not reorganize around the new goal, only shifting slightly in its direction. No such shift was seen in the place fields of non-ConSink cells (Extended Data Fig. 14d).”

7. Line 251 – “the centre of the environment” – the egocentric coding toward centre of the environment was also shown earlier by Wang et al. 2018 from the Knierim lab, but they might mention that this measure can be ambiguous relative to coding of environmental boundaries as shown in recordings of retrosplenial cortex by Alexander et al., 2020.

Response: We have now added the Wang et al. reference to “the centre of the environment”, and also mentioned that coding to environmental boundaries can appear as coding to the centre (line 226):

“...the centre of the environment^{26,27} (though egocentric coding to environmental boundaries can appear as coding to the centre²⁸)”

8. Line 346 – Is there a difference in calculation technique between the average sink position (black dot) in Fig. 1h and the average ConSinks shown with open circles in Fig. 2a? Perhaps the difference is only due to this being different experiments. But Figure 1h itself labels the black dot as “average sink position” whereas the figure legend says “centroid” so perhaps this indicates a difference.

Response: There is no difference in technique between Fig. 1h and 2a. In Fig. 1h, the average sink position has been changed to an open circle of the same colour as the individual sinks for consistency with Fig. 2a, c, and e. We have removed the reference to centroids in the figure legend, since they are labelled in the figure already as “average sink position”.

9. Line 348 – Fig. 1i “average vector fields” – are these the vectors for individual ConSink place cells or the average across place cells for a location? It is confusing whether or not this is based on all cells firing in a location because the vector fields are shown on a cell by cell basis for significant ConSink cells in Figs. 2g and 2h.

Response: The plots shown in Fig. 1i and Fig. 2g and h are averages constructed from all ConSink cells. For each ConSink cell, a platform specific vector is calculated with heading equal to its mean head direction and its length equal to its MRL multiplied by its firing rate. These vectors are summed across cells within each platform, and the headings of these summed vectors are plotted as arrows. The text now reads (line 60):

“The population vector fields (calculated by summing platform-associated heading vectors across cells) for each animal ...”

A paragraph detailing the methodology has been added to the Methods (also included below in response to referee 2, comment 5; line 569):

“For population vector fields, bins instead corresponded to maze platforms. For each cell on each platform, we calculated the cell’s mean allocentric head direction. We then created a unit vector from this head direction value, and scaled it by multiplying it by both its platform-associated mean firing rate and MRL. Finally, these scaled vectors were summed across all ConSink cells which fired spikes on the platform, and a direction was calculated from the resulting vector. The population sink position was calculated using the same search across xy positions as for individual cells, converting each platform associated head direction to a relative direction whose contribution was scaled according to the length of its associated vector, and calculating the MRL, taking as the sink the position with the maximal MRL value.”

10. Line 392 – What is the thin black line in Figure 4L? Is it the average (fantail) of the three rats? If not, could they plot the fantail here directly for comparison?

Response: The average fantail of the 3 rats is shown in k (now Figure 3). The black line in l corresponds to k, with the one difference being that it averages the negative and positive directions (e.g. the black line at 30deg is the average of the fantail at -30deg and 30deg). This has been clarified in the legend (line 376):

“l, Population vectors for each animal conform to this model. Individual points correspond to the fantail vectors from each animal; note that positive and negative directional (e.g. -30° and 30°) vectors are averaged. The black line is the average across the 3 animals.”

11. Line 466 – “ In the second recording session.” – This phrase appears twice .

Response: This has been corrected (line 457).

12. Line 467 – “To goal 1” – this reads as if it is the same goal used in the first recording session (which was the assumption I made based on the main text). If not, it should be emphasized here AND in the main text that goal 1 on the shift goal task is NOT the same as the initial goal used in Figure 1. Maybe it would be better not to call it goal 1 as that suggests it is the first goal in the overall experiment, whereas it might just be the first goal in the two goal task?

Response: This has been corrected, as detailed above.

13. Line 576 – “all animals ran 13 trials to goal 1” – this is still confusing as it suggests goal 1 is the goal in Fig. 1, but the goal locations don’t seem to match between Fig. 1 and Fig. 2.

Response: This has been corrected, as detailed above.

14. Line 715 – it is hard to see the rat in Extended Data Fig. 2. Is there a chance that an outline could be drawn or a line could be drawn down the midline of the rat to ease viewing. It is also very hard to tell that it is on a raised platform, which is surprising as it suggests the camera aligns the borders of the raised platform perfectly with the lower platforms.

Response: We agree it is hard to see the rat, and a line down the midline is a good suggestion. In panel a, however, we mainly want to draw attention to the path of the rat (in white), and putting a line that is large enough to be visible detracts from this. However, in panel b, where we want to show the directional sampling of the rat, we have added 3 coloured ellipses over the animal’s cervical (yellow), thoracic (green), and lumbar (blue) regions to show his position and orientation; it will be necessary for the reader to zoom in on the images to see the ellipses clearly, but we did not want to obscure the rat itself. We have added text (line 731):

“3 ellipses are drawn over the animal’s cervical (yellow), thoracic (green), and lumbar (blue) regions to show his position and orientation.”

In terms of being able to see the raised platform, this depends on the exact location of the platform on the maze. For a platform directly below the camera, the vertical sides of the platform will not be visible. However, towards the maze periphery the sides facing towards the maze centre are quite visible if one is accustomed to looking at these images, e. g. in the first panel in a, 3 platforms are raised on the left, lower side of the maze, and the vertical sides are visible to the right (see that the raised platforms appear quite distant from the adjacent lowered platforms). We have now included a video shot from the overhead camera (Extended Data Video 1); the movement of the platforms in the video makes it easier to see their vertical position.

Referee #2 (Remarks to the Author):

This paper investigates an important question in the study of navigation, addressing the mechanism and computations used by the hippocampus to support goal-directed navigation. Using their newly developed honeycomb maze, the authors demonstrate that CA1 place cells display egocentric vector coding between the animal’s current location and the goal location. The authors further show that, in principle, all of the information that the animal needs to guide its behavior to the goal at every step is contained in this vector code.

Although the demonstration of egocentric coding in place cells and other cells in the medial temporal lobe system is not novel, the authors relate this coding to behavior in an interesting way. Their conclusions about vector coding are important. However, there are a number of concerns with the data analyses that weaken the case, to the point that it is not clear whether their conclusions are supported strongly enough. These concerns would have to be addressed in order to make the case compelling enough for publication.

Major

1) p. 3, "Of these, 77 (29%) displayed firing patterns within the place field which were best described by vector fields converging on a location that, following vector field notation, we term a convergence sink." I assume that this is a comparison with allocentric head direction tuning. However, the authors comparison between the egocentric coding and allocentric HD coding (Extended Data Fig. 5b) appears biased to favor egocentric coding. For each cell, multiple egocentric bearing tuning curves are calculated, one for each of the 986 (34 x 29) potential ConSinks. The egocentric curve with the best ConSink is chosen. However, there is only one allocentric HD tuning curve (by definition). Assuming that there is some amount of noise and stochasticity in the firing, a place cell with allocentric HD tuning (e.g., perhaps Cell g of Figure 1) would show a well-tuned curve with any of the more peripheral ConSinks, and the algorithm would pick the best one out of many (ie, the extreme tail of the distribution). Because of this biased selection, chances are high that the best egocentric tuning curve will be tighter than the single allocentric tuning curve. The authors need to make a fair comparison between the ConSink-based tuning curves and the allocentric tuning curve to state with certainty how many of the apparent ConSink cells are actually allocentric HD cells (or cannot be statistically distinguished from HD cells). It is not readily apparent to this reviewer how to do this, but it is a critical component.

Response: Our identification of ConSink cells does not rely on a comparison between a given cell's allocentric and egocentric (relative to the sink) tuning. Rather, it depends on a comparison between the strengths of egocentric tuning using the actual spike positions and shuffled positions to ask whether the spatial distribution of spikes is arranged optimally for egocentric tuning. Nevertheless, we include a comparison of allocentric and egocentric coding in the ConSink cells as evidence in support of our interpretation that these cells encode egocentric direction (i.e. the panel that now comprises Extended Data Fig. 7a, reproduced below), and thus the reviewer has raised an interesting and important point.

To examine this issue, we ran a simulation in which, for each of the 77 ConSink neurons from Session 1, we assigned new head direction values for each spike. These head directions were calculated by using a given cell's true mean allocentric head direction and adding increasing levels of noise. We then calculated both allocentric tuning and egocentric tuning (i.e. the maximum MRL calculated across all possible sink positions). The curve in panel b shows that the egocentric MRL closely

tracks the allocentric MRL at all noise levels, and becomes relatively larger at the highest noise levels (c, d). However, at these higher noise levels, the egocentric MRLs are lower than in any of the cells we identified as ConSink cells (see panel a; the lowest egocentric MRL was ~ 0.2). Only at a noise level of 100deg standard deviation does the simulation produce relatively larger egocentric MRLs that are within the range of values that we actually observed; this led us to ask whether the difference in egocentric vs allocentric MRL values for the ConSink cells with the lowest egocentric MRLs was significantly greater than that of cells with 100deg noise in the simulation. We examined all ConSink cells with egocentric MRLs less than either 0.4, 0.3, or 0.25. In all 3 cases, the increase in the egocentric MRLs relative to the allocentric MRLs were greater than could be explained by noise in an allocentric signal (panel e).

We have added this data to Extended Data Figure 7.

We have added to the following figure legend (line 811):

“Extended Data Figure 7 | Tuning to the ConSink is not an artefact of allocentric tuning in ConSink cells. a, MRLs calculated from allocentric (allo.) and relative (rel.) head directions for each ConSink cell during task. Note that for every ConSink cell, the MRL calculated from relative directions is greater than the MRL calculated from allocentric head directions. b-e, Noise in a purely allocentric head direction signal can't explain the greater egocentric MRLs in ConSink cells. b, We simulated purely allocentric cells by assigning new head direction values to the spikes fired by each ConSink cell. These head directions were calculated by using a given cell's true mean allocentric head direction and adding increasing levels of noise. We then calculated both allocentric and egocentric MRLs as was done for our real data. In these simulated allocentric cells, egocentric MRLs closely tracked allocentric MRLs, and both decreased with increasing levels of head direction noise. c, Only at noise levels above 100deg standard deviation do egocentric MRLs become larger than allocentric MRLs. d, The MRLs at 120deg noise and above were smaller than any we observed in our identified ConSink cells, and thus irrelevant. e, Only in cells with ~100deg of head direction noise, and therefore MRLs of ~0.2 length, would we expect true allocentric cells to have relatively larger egocentric MRLs within the range of values we observed in our data. However, the differences in egocentric MRLs relative to allocentric MRLs in our most weakly tuned ConSink cells were still much greater than could be explained by noise in a purely allocentric signal.”

2) I would suggest that the authors do away with the terminology of a “ConSink cell.” There is a proliferation of names that identify particular properties of cells with a classification of a new cell type. Although this makes sense when describing clearly different types of cells (eg., place cells, grid cells, head direction cells, border cells, etc.), it adds confusion when a newly discovered property of these cell types is called a new cell type. Furthermore, in these other cases, the term reflects the putative representational function of the cell (e.g., a place cell represents a place in the environment, a grid cell represents a grid-like coordinate frame, etc.). What does a ConSink cell represent? The ConSink? I don't believe that the authors think this. (Similar concerns apply to other authors' usage of the term “anchor cell” in Reference 12 to refer to a very similar property as that describe in the present paper.)

Response: We agree that these cells may not be different from place cells and apologize if our nomenclature suggested otherwise. We use the term ConSink cell as a shorthand for ConSink place cell, i.e. the subclass of place cells which have statistically significant ConSinks under the circumstances of the experiment. In fact,

our data suggests that all place cells may be able to encode relative direction to a location in the environment (i.e. the goal switch data in which some cells have ConSink tuning during Goal1, and others during Goal2). We would like to keep the term for conciseness but have added “ConSink place cells” on page 3 (line 62) where the concept is first introduced as well as in figure 1 (line 331), and have added the following text in the discussion (line 230):

“It must be left to future studies to determine how these Convergence Sinks are created and manipulated, to elucidate the properties of the underlying reinforcement mechanism, and to determine whether ConSink cells represent a distinct cell type or whether ConSink tuning is simply a property place cells can turn on and off.”.

We trust that these changes will make it clear to the reader that we are not suggesting a new class of cell in the same way that identifying the directionality of a place cell in a particular context (viz directional place cell) does not convey the idea that this is a new class of cell or even a permanent property.

3) P. 4: “Similarly, the vector fields and their associated population ConSinks moved from the original goal locations towards the new goals.” This is clear only for Rat 3. Rats 1 and 2 do not show an obvious shift toward the goal. The apparent shift could be a statistical artifact; that is, given that the average ConSink is initially near Goal 1, assume that the average shifts to a new, random location for Goal 2. Because $\sim 2/3$ of the platform encompasses locations closer to the new goal rather than farther away from it (relative to the initial position), the analysis is biased in favor of this effect. The authors need to estimate the probability of the vector field average goal shifting closer to Goal 2 by chance and use this as the null hypothesis.

Response: In considering the comment, we re-examined the relevant panels (Fig. 2g and h). During the goal 2 epochs in Rat 1 and 2, we felt that the vectors seemed to be pointing towards the goal 2, yet the population sink positions didn’t reflect this. In the original calculations for the sink position, we had simply used all spikes fired by ConSink cells, whereas to create the vectors, we had instead calculated platform-associated vectors for each ConSink cell and then summed these vectors across cells. We therefore have now recalculated the sinks using these vectors (see Methods, line 569). We scaled the contribution of each platform-associated vector by its vector length. Using this method, the sink positions in both Rats 1 and 2 moved closer to the goals. We also examined the MRL values at the Goal 1 and Goal 2 positions, and found that the MRLs at Goal 2 were higher than at Goal 1 during the

Goal 2 epoch in all 3 animals (and vice versa during the Goal 1 epoch); thus both methods show consistently that the ConSink population is more tuned to Goal 2 during the Goal 2 epoch in all animals (and to Goal 1 during the Goal 1 epoch). Also, note that as we explain above, Rat 2's goal 1 in this figure is actually the second goal on which he was trained; that the population sink to Goal 1 in b is overtop the goal supports our interpretation that the sink moves to the goal with training.

a, Left, Population field vectors and sink position during Goal 1 (top) and Goal 2 (bottom) for Rat 1. Right, Maps of MRL values during Goal 1 (top) and Goal 2

(bottom). b, c, as in (a) but for Rat 2 and 3. d, The MRL values taken from the MRL maps at the goal positions. Note that the field vectors and summary panel d are presented in Fig. 2, and the MRL maps presented in Extended Data Figure 11.

The figure legend for Figure 2 panel i (referred to as d, above) is (line 357):

“i, The MRL values taken from the MRL maps (shown in Extended Data Figure 11) at the goals during both epochs. Note that MRL values are always higher at the current goal than the subsequent or previous goal.”

The figure legend for Extended Data Figure 11 (which contains the MRL maps above) is (line 876):

“Extended Data Figure 11 | MRL maps calculated from population vector fields show stronger tuning to the current goal than the subsequent or previous goal. MRL maps during Goal 1 (top) and goal 2 (bottom) for rats 1, 2, and 3 calculated from vector fields shown in Fig 2g and h (these maps are analogous to the single cell MRL map shown in Extended Data Figure 5b). The MRL values shown in Fig. 2i are taken from the goal-centred locations in these maps.”

4) P. 10: The authors state in the Discussion that “the animal simply needs to move in the direction of highest firing rate afforded by the choices available, essentially comparing the lengths of the available branches of the Fantail.” How is this done computationally based on these results? Does the animal do a VTE behavior?

Response: Yes, the animal engages in scanning behaviour before making his choice. We have added a figure (Extended Data Fig. 3a) that shows the animals perform, on average, a nearly full 360° rotation before making their choice and moving onto the next platform; Extended Data Fig. 2 shows an example of this behaviour. Extended Data Fig. 17 shows that prior to subsequent “correct” choices, the animal samples head direction relatively broadly both before the choice platforms come up (Wait Period 1) and as they come up (Wait Period 2). Where the actual decision occurs is likely in downstream structures which could be the subject of future studies.

Can the authors see these differences in population firing rate?

Response: Yes, this is shown in Figure 3k, and also in correct and error trials in Fig. 4d. The text has now been clarified (line 131):

“Although we have represented the mean allocentric direction of population spiking in the vector fields (see Fig. 1i), the underlying data can also be represented as a set of vectors whose lengths represent the population firing rates in the corresponding directions relative to the goal, and whose average points towards it (Fig. 3k).”

The conclusions of this paper would be much more compelling if the authors could show evidence of this comparison taking place while the animal was deliberating its choice. Without this analysis, the conclusions are much weaker and uncertain.

Response: Extended Data Fig. 17 shows that the animals broadly sample head direction during the period before the platforms start to rise (Wait Period 1), as well as during the time after the platforms start to rise but before the animal makes his choice by moving to a new platform (Wait Period 2). Together with Fig. 4, these data show that on correct trials during the time when the animal is waiting for the next set of platforms, he samples all directions and the firing rates decrease with deviation from the goal direction (Fig. 4d, left). We have added an explicit statement to this effect in the text (line 171):

“During the wait periods, the rat systematically scans the environment (Fig. 4a, b; Extended Data Fig. 3a), providing an opportunity to sample the different branches of the fantail.”

5) Figure 1i: How are the average vector fields calculated? Summing all spikes from a recording and dividing by time? Calculating a vector for each cell and location and calculating the mean vector? Or some other algorithm? (Apologies if this is in the Methods but I missed it.)

Response: We calculated a vector for each cell on each platform and summed the vectors across all ConSink cells. The vectors were calculated by creating a unit vector whose direction was the mean direction of the cell on the corresponding platform, and then scaling the vector by the mean firing rate multiplied by the MRL of the cell's allocentric HD distribution. We have added to the Methods (line 569):

“For each cell on each platform, we calculated the cell's mean allocentric head direction. We then created a unit vector from this head direction value, and scaled it by multiplying it by both its platform-associated mean firing rate and MRL. Finally, these scaled vectors were summed across all ConSink cell's which fired spikes on

the platform, and a head direction was calculated from the resulting vector, which is represented by each arrow. The population sink position was calculated using the same search across xy positions as for individual cells, converting each platform associated head direction to a relative direction, and calculating the MRL, taking as the sink the position with the maximal MRL value.”

6) The finding that the mean direction of significant ConSinks was indistinguishable from 0 suggests that these cells are really just place cells that fire most strongly when the rat is pointing toward the goal location, not a more general vector coding scheme. Related findings are found in Acharya et al. Cell (2016) from Mehta’s group, in which directional firing toward cue locations was revealed in their VR system that disrupts the spatial firing of place cells. The same inputs to pyramidal cells that produces this type of cue-directed directional tuning may also produce goal-directed firing in place cells in the present task.

Response: We would argue that the results show that the ConSink/fantail representation is much more than "...place cells that fire most strongly when the rat is pointing towards the goal location...". As we have shown, many of the individual ConSink cells are organized around locations which are not at the goal but at the population level do point to the goal. Equally importantly, the firing in other non-goalward directions is organized to provide information about the value of those other directions, providing for a powerful flexible navigation system. Results of the Hardcastle LN analysis suggests that vector fields contain more than directional information and provide the true distance and direction vectors required for non-goalward navigation. We thank the referee for reminding us of the Acharya et al reference and have added it in the following text (line 144):

“Place cell firing is omnidirectional during random foraging in an open field with walls⁶ (although see^{7,20}), but polarized or unidirectional in linear environments^{9,21}, virtual reality²⁰, and in open fields after goals are introduced²².”

7a) p. 29: A shuffling procedure is not technically a bootstrap. A bootstrap is a specific procedure in which an underlying population distribution is estimated by multiple iterations of random sampling with replacement of the same number of observations as in the data sample. This is not what the authors performed here.

Response: We have corrected our description of the shuffling procedure to omit the term “bootstrap”. The text now reads (line 529):

“To test whether a cell was significantly tuned to direction relative to the candidate sink, ~~we used a bootstrap method in which~~ we shuffled the cell’s head directions

such that the head directions were no longer associated with their actual positions on the maze.”

7b) In addition, please provide more details of the exact shuffling methods used. These details are necessary for the read to evaluate the validity of the shuffling procedure and whether it is a reasonable representation of the null distribution based in the experimental hypothesis. Were the head directions assigned randomly? If so, how does this change the temporal statistics related to the animal’s head direction behavior and does this affect the validity of the shuffling procedure? Or was the train of head direction measurements shifted relative to the train of spikes, or some other method?

Response: Yes, head directions were assigned randomly. To address whether any ConSink cells’ apparent egocentric tunings were an artefact of the temporal structure of their spike trains, we carried out an additional procedure where we shifted spike trains by random amounts, and recalculated their sink centred MRLs using the shifted values for HD and position. Any cell’s whose previously calculated MRL fell below 95th percentile of the shifted distribution was removed from the pool of ConSink cells; this resulted in removal of 1 cell from the open field foraging data, and 1 cell from the goal switch data. Details of the new shift procedure have been added to the methods.

We have added the following text to the Methods (line 537):

“To account for the disruption of the temporal structure of spiking caused by our shuffling procedure, we performed a second test in which we shifted the spike trains in time (minimum shift of 60 sec), recalculating the strength of tuning to the sink position using the shifted positional and head direction values. A control distribution of MRL values was created from 1000 repeats of the shifts, and cells were discarded if their real MRL was less than the 95th percentile of this control distribution.”

Minor

8) p. 1: “The distribution of these convergence sinks was centred near the goal location,...” This is an overstatement, as it convincingly true in only 1 of 3 rats. (FROM LINE 15)

Response: This sentence is from the summary, and refers to the data from Session 1 and the Goal1 data from Session 2 (note that when we refer to the goal switch data in the next sentence, we are specifically referring to the Goal2 data). While in Session 1, the average ConSinks are merely near the goal, in Session 2, in Rats 1

and 3, they are at Goal1 itself (and in Rat 2, quite near Goal1), so the data indicates that “near the goal location” is not an overstatement.

9) p. 2: (but see 8)” The authors should also cite Rubin et al. J Neurosci 2014 from the Ulanovsky lab
Response: The citation has been added (line 30).

10) p. 2: ref 10—The authors should also cite Kobayashi et al J Neurophysiol 1997 from Ono lab.
Response: The citation has been added (line 31).

11) p. 2: “human” The Kunz paper is not from dorsal hippocampus (or at least not mainly from the corresponding region in humans, ie, posterior hippocampus).

Response: We have omitted the word “dorsal” on line 32:

“Recent work has shown that cells in the ~~dorsal~~ hippocampal formation encode heading towards a goal and other locations in the bat¹², mouse⁷, primate¹³ and human¹⁴.”

12) p. 4: “(282 CA1 place cells; Rat 1: 93 cells; Rat 2: 96 cells; Rat 3: 93 cells)” This seems very strange to have almost the identical number of cells for each rat. Is there something about the automated sorting method that sets a number of cells to identify, regardless of the isolation quality?

Response: The numbers of cells do not result from the automated sorting method, which generated many clusters that had to be manually removed, merged, or split. One possible reason for the consistency in cell number could be the relatively large number of tetrodes (32) we used; if variability in tetrode quality is a factor in determining the number of cells recorded, having more tetrodes per animal should result in more similar patterns of electrode quality.

Also, note that the numbers listed are for cells that met the criteria in either Goal 1 or Goal 2, and in fact, there is a little more variation if you look at those numbers separately: for Goal 1, the numbers are 87, 88, and 81, and for Goal 2, the numbers are 87, 95, and 93. The numbers in session 1 are also quite consistent but less so than in the goal switch (session 1: 89, 94, 83), and taken together show that there is somewhat more variability than the statement you quote indicates.

13) p/ 4: “32% (83/256) of principal cells during the original” Do the authors mean “place cells” rather than principal cells? For Fig 1 they said that 29% of “place cells” had ConSink tuning

Response: We meant place cells. Note that the numbers have changed slightly (see response to comment 16, below). The text has been corrected to read (line 78):

“...and 33% (84/256) of place cells during the original and 23% (62/275) during the shifted-goal navigations...”

14) p. 6: Extended Data Fig. 10c: The relative direction appears to represent 180 degrees less than the other directions. Is this significant?

Response: Yes, it is significant. This is the expected result, given that ConSink cells had a mean preferred relative direction to the sinks of ~0deg (see Fig. 1j). The following text has been added to the figure legend (Extended Fig. 15, line 942):

“Note that the circular distribution in (e) is significantly non-uniform ($p < 0.001$), as expected (see Fig. 1j).”

15) p. 10: “Finally, we find that firing towards the goal or the individual sinks is reduced and the fantail pattern altered on error trials, indicating that ConSink cells are crucially involved in navigation the goal.” This is an overstatement that implies causality, which has not been demonstrated in this paper. Although the data are highly suggestive, they do not demonstrate that these cells are “crucially involved.”

Response: We have changed the text to read (line 200):

“Finally, we find that firing towards the goal or the individual sinks is reduced and the fantail pattern altered on error trials. Thus, ConSink cells exhibit properties that could support navigation to the goal.”

16) Figure 2a, 2c: There appear to be fewer dots than indicated by the total number of cells mentioned in the text.

Response: We mistakenly included the wrong n value in the text. These numbers were inadvertently changed from the correct numbers during manuscript preparation, and did not accurately reflect the numbers used in the analysis. The numbers should have been 85/256 and 62/274 (not 83/256 and 70/275 as they were in the original text); note that there is now one less cell during Goal1 because of the shift analysis detailed above (i.e. 84/256). There was also overlap of some points preventing all from being visible; these have now been jittered to make them visible. The text has been changed to read (line 78):

“...33% (84/256) of place cells during the original and 23% (62/275) during the shifted-goal navigations...”

17) P. 24: Maze section. Please describe how the animal's behavior was shaped initially.

Response: We have added the following text to the Methods (line 424):

“Animals were initially trained to consume the food reward (honey-flavoured corn flakes) in their home cage. Once they were consuming the food in their home cage, they were brought onto the maze, placed on the reward platform, and given food reward. Once they were consuming the food on the reward platform without hesitation (1 or 2 days), we began to run task trials, initially running small numbers of trials, and increasing the number gradually in preparation for the recording sessions.”

18) P 28-29: last 2 lines of p 28 and first 3 lines of p. 29 are not clear. What does it mean that “the distributions were summed according to the number of spikes the cell fired on each platform”? The next sentence also needs clarification as to how this analysis takes into account uneven sampling.

Response: This paragraph refers to the correction of a candidate cell's distribution of relative directions by the relative direction “occupancy” (i.e. the amount of time spent by the animal facing in various directions relative to the candidate sink position).

Because a given cell might only fire at certain locations on the maze, and the animal's directional occupancy might change across positions, for each cell, we calculated an occupancy (referred to as control) distribution based on the directional occupancy on those platforms where it fired its spikes. This required a directional occupancy distribution for each platform. For a given cell, the directional occupancy distribution for each platform was multiplied by the number of spikes the cell fired there. These distributions were then summed across all platforms on which the cell fired spikes. Finally, the “raw” directional distribution of the cell's spikes could be divided by this control distribution in order to correct for uneven sampling.

The text has been altered to read (line 517):

“For each cell, the distributions were scaled according to the number of spikes the cell fired on each platform. After scaling, the distributions were summed across platforms. Finally, the cell's relative direction distribution was divided by the summed control distribution, providing a corrected distribution taking into account any uneven sampling of relative direction by the animal across those positions at which the cell fired spikes.”

19) p. 29, last 3 lines: Was this procedure also performed for the allocentric HD cell plots for comparison?

Response: Yes, it was, but using allocentric head direction rather than relative direction. This has been clarified in the Methods (line 527):

“Note that the same correction was also performed for calculation of allocentric tuning.”

20) Extended Data Fig 7e (caption): “indicating stability of sink position within this sub-population across the goal switch.” I wouldn’t call this a good test of stability per se—it just indicates that the sink positions are not random.

Response: We have altered the text to Extended Data Fig. 9 legend (line 858):

“Together, (e) and (f) indicate that sink position is not solely dependent on the goal.”

Referee #3 (Remarks to the Author):

Ormond and O’Keefe record from dorsal hippocampal place cells while rats perform a goal-directed task on a maze in which navigation is constrained to stepwise decisions to move in one of two candidate directions. They report that, in addition to coding allocentric position in the environment, place cell activity reflects egocentric bearing to vector field convergence sinks (here dubbed ConSinks) which, at least when averaged together, are located near goal locations. Many such ConSinks were located far from the goal. Relative bearing tuning for goal locations was reduced during free foraging and on incorrect trials. The authors argue that the dual computation of egocentric orientation to a goal, allocentric orientation (relative to the recording room environment) and spatial position enables the hippocampus alone to solve the task. That is, the offset between the egocentric and allocentric orientations at any given location could be used to compute an optimal reorientation of the animal in the direction of the goal. The actual computation would, presumably, be carried out by a structure receiving CA1 output. Notably, one would have to assume that such a structure would determine the optimal reorientation from a large number of different CA1 ensemble patterns as these differ most robustly by environmental location. A number of related findings regarding goal-related directional modulation of place cell firing (Markus et al., 1995; Acharya et al., 2016; Sarel et al., 2017; Gauthier et al., 2018) have already been reported both in the hippocampus and parahippocampal areas (Wang, Chen, et al., 2018). Because of this, the true novelty of the work lies in providing the experimental context of an actual navigational behavior and a full-fledged consideration of a mechanism for it (combining location, distance to goal, egocentric orientation to goal, and allocentric orientation). Below are specific comments/concerns.

1) There should be considerably more analysis regarding the behavior of the animal on the task and how it relates to the convergence sinks observed.

For instance, the authors claim at multiple points that the animal frequently samples all head directions while waiting on each platform but there is no explicit analysis of this.

Response: We have added a plot showing the directional range covered by the animal per choice on the maze from the time when the platforms begin to rise to the

time when the animal moves to the next platform (Extended Data Fig. 3a). Note that ranges greater than 360deg indicate that the animal continued to scan in the same direction (i.e. multiple rotations); if an animal scanned 360deg and then counter-rotated back to the starting direction, range is calculated only as 360deg. From the distribution of all ranges, it is clear that the animal covers most directions during most choices. The median range covered per choice is 276 degrees and the mean is 265 degrees. We have also added a plot showing total allocentric directional occupancy and goal direction occupancy for each animal (Extended Data Fig. 3b).

2) How does the animal's time to choice, egocentric bearing, allocentric heading, etc. vary as a function of distance to the goal?

Response: The animal's time to decision decreases as he gets closer to the goal. However, the distributions of allocentric and relative-to-goal directions don't show any clear systematic change with goal distance in any of the 3 animals. These data are presented in Extended Data Fig. 3c and d.

The figure legend for Extended Figure 3, which combines the data from Comment 1 and 2 above, reads (line 737):

“Extended Data Figure 3 | Rats sample a large range of possible directions while navigating the honeycomb task. a, Histogram showing the directional range covered by the animals per choice from the time when the platforms begin to rise to the time when the animal moves to the next platform. Note that ranges greater than 360° indicate that the animal continued to scan in the same direction (i.e. multiple rotations); if an animal scanned 360° and then counter-rotated back to the starting direction, range is calculated only as 360° . b, Allocentric (allo.) and relative-to-goal (goal) directional occupancy for each animal. Note that goal direction is not oversampled. c, The time the animals take to make their decision decreases as they get closer to goal; however, this does not seem to prevent them sampling the full range of direction at short-goal distances (d).”

3) How does the egocentric bearing to the goal or overall directional heading vary over learning of the goal location and how does it change when a new goal is introduced?

Response: Unfortunately, we don't have this for session 1, since the animals were pre-trained. When the new goal is introduced in the goal switch session, there is no clear discernible pattern across animals (below). Rat 1 seems to be oriented towards the new goal after the initial switch, and becomes less so with additional trials, whereas Rats 2 and 3 show an original tendency to orient away from the new goal which then becomes less so with additional trials.

The full dataset (266 cells with significant spatial information scores) seems rather small as is the number of neurons exhibiting a ConSink (77).

Response: That is only half the data set. The Goal Switch data includes another 121 ConSink cells (59 during Goal 1 only, 37 during Goal 2 only, and 25 during both, from 274 place cells). Thus, our study includes a total of 198 ConSink cells, identified from a larger population of 540 place cells.

4) This issue with the work is exacerbated by the fact that the test for directional tuning has a rather low threshold for significance (resultant length vs that for a distribution of resultants following randomization of orientations relative to spikes).

Response: We have added an additional test of significance taking into account the temporal structure of spiking, as requested by Reviewer 2, Comment 7b.

5) Similarly, the dataset is constrained with respect to the total number of trials (13-26) in any given recording.

Response: While the number of trials may appear small, the sessions were in fact quite long (minimum 2hrs), and we arrived at these numbers of trials because they appeared to be the maximal number of trials any well-trained animal would run in one session; note that these numbers of trials required between 100 & 190 individual choices for each animal (see Extended Data Fig. 1 and 8). We anticipate that new recording technologies (i.e. multi-shank Neuropixel probes) will allow us to track neurons across sessions, allowing us record individual neurons over much larger numbers of trials in the future. While this might give us a finer grained analysis of some of the variables, we do not think it will significantly alter the overall picture given that the results are clear with the present number of cells and trials.

6) The distribution of ConSinks in all 3 rats may, when averaged, reflect tuning to the goal locations. However, their spatial distributions are so broad that it seems implausible that an ensemble of CA1 neurons provides more than a 'lean' to the relative direction and distance to the goal. To be convincing as a mechanism actually guiding navigational behavior, one would hope for a far less noisy signal.

Response: In most real-world navigation tasks the direct direction to the goal is not always available and the job of the hippocampus is to construct a set of vectors which support navigation in the absence of this pathway. This would explain why there are ConSinks which are not located close to the goal since one of the functions of this system appears to be to evaluate the valence of all possible directions. This is

supported by the observation that the non-goal directed vectors in the fantail are not random as one would expect if they were noise but are systematically organized as they fall off from the direction of the goal.

7) If a neuron has a punctate spatial receptive field (i.e. is a place cell) and the directional sampling is constrained (e.g. by more often facing the goal location) then one could imagine a scenario in which ConSinks emerge independent of actual directional sensitivity. This may relate to the overrepresentation of relative bearings directly in front of the animal.

Response: Taking the directional occupancy from the first version of the manuscript, together with the new occupancy data we provide (see above, in Response to Referee 3, Comments 1-3), shows that there is perhaps a small oversampling of goal direction in Rat 1, but certainly not in Rat 2 or 3, so we think it very unlikely to account for our results. Furthermore, we correct for directional bias in each cell, not across the whole maze, but specifically for the directional occupancy at those locations at which a given cell fires its spikes (see also response to Referee 2, Comment 18).

8) For this reason, the authors should consider downsampling to match head directions on each platform and rerunning a number of their analyses.

Response: To address this comment, we downsampled spikes for each ConSink cell fired on each platform according to the allocentric directional occupancy on that platform. We then compared each cell's newly identified sink position and preferred direction to its previously calculated value, as well as to all other values previously calculated in the rest of the ConSink population. We found that sink positions were more similar within cells than across cells (Extended Data Figure 5d), and x and y coordinates were strongly correlated across the two techniques (Extended Data Figure 5e, f). Similarly, preferred relative direction was more similar within cells (panel g). The strength of tuning was also highly correlated between the two techniques (panel h). Lastly, we found that our technique did not overestimate the strength of tuning, as tuning was slightly, but significantly, stronger in the downsampled data (Extended Data Figure 5i).

The figure legend has been added to (line 773):

“d, To confirm the validity of our calculations, we recalculated the sinks in our identified ConSink cells using a downsampling method, in which, for each cell on each platform, the spikes were downsampled according to the directional occupancy in allocentric coordinates (see Methods). We then compared the distances between the sinks calculated with the 2 different methods (our “divide by scaled occupancy” method, and “downsampling” method) within and between cells. We found that sink positions were more similar within cells than across cells, and x and y coordinates were strongly correlated across the two techniques (e, f). g, Similarly, preferred relative direction was more similar within cells. h, The strength of tuning was also highly correlated between the two techniques. i, Lastly, we found that our technique did not overestimate the strength of tuning, as tuning was slightly, but significantly, stronger in the downsampled data.”

We have added to the Methods (line 559):

“To confirm the validity of our methodology, we recalculated sink positions and preferred directions using a downsampling method (see Extended Data Fig. 5d-i). For each cell on each platform, the spikes were binned according to allocentric head

direction (24 bins of 15° width). Spikes within each bin were then downsampled according to the total directional occupancy in that bin; i.e. if the animal spent relatively more time facing a particular direction, the spikes fired in that direction were downsampled by a proportionate amount. For a given cell, spikes were then summed across platforms and the sink position, preferred relative direction, and strength of tuning (MRL) were calculated. This was repeated for each cell 1000 times, and the mean values calculated.”

9) Use of the Hardcastle model alone may not account for oversampling of specific headings which could cause considerable covariance of predictors related to egocentric goal bearing and allocentric heading or spatial position. Further, the GLM analysis relating to Figure 4 should be run on all cells, not just those possessing ConSinks. If place cells without ConSinks have significant sensitivity to the relative direction, allocentric direction, or distance to the goal what would that indicate about the presence of ConSinks and the subsequent conclusions?

Response: We included this analysis to show, in a quantifiable way, that ConSink cells are not tuned solely to egocentric direction, but also to additional variables that could, together, support navigation on the maze. We would predict that many non-ConSink cells could yield positive results for 3 reasons. First, as we diagram in manuscript Fig. 4b-e, tuning to distance and allocentric direction to sink are sufficient to produce a place field, so many “typical” place cells could show tuning to these variables at certain sink positions. Second, as Reviewer 2 pointed out, purely allocentric directional tuning can lead to relatively large mean resultant vectors when calculating egocentric tuning at certain positions within the environment. Third, it is possible that some proportion of ConSink cells simply didn’t reach significance in our identification procedure.

Running this analysis on all non-ConSink cells (firing a minimum of 500 spikes during the task; n = 189) showed that, while many non-ConSink cells were significant for 1 of the models, significantly fewer were positive compared to the ConSink cell population (panel a; see manuscript Fig. 4; 86% of ConSink cells, 62% non-ConSink cells, Chi squared test, $p < 0.001$). Further, a greater proportion of ConSink cells were significant specifically for models including relative direction as a variable (82% vs. 46%, Chi squared test, $p < 0.001$). The strength of egocentric tuning (i.e. MRL length) was significantly weaker in non-ConSink cells that tested positive in this

analysis as compared to ConSink cells, providing an explanation for why these cells tested negative in our initial identification (panel b).

10) What happens if you run this model with bearing and distance calculated to any non-goal zone on the arena?

Response: To test this idea, we chose 3 other positions to use as control sinks in the Hardcastle model. First, we took the position of least tuning (i.e. MRL minimum; see inset panel b, which is taken from Extended Data Fig. 5b, blue position), second, the sink position reflected across the x and y axes (opposite to sink), and third, the farthest distance in the field of view from the sink position. The model provided significantly less information about a given ConSink cell's spiking when using any of these positions rather than the identified sink position itself (panel c).

This data has been added to Extended Data Fig. 15. The figure legend now reads (line 930):

“b, To test the validity of the LN model results, we recalculated the log-likelihood (LLH) increase in spike prediction using 3 non-sink control positions: first, we took the position of least tuning (i.e. MRL minimum; see inset, which is reproduced from Extended Data Fig. 5b, blue position), second, the sink position reflected across the x and y axes (opposite to sink), and last, the farthest distance in the field of view from the sink position. c, We calculated the difference between the LLH increase using the real sinks and calculated using these 3 control sinks. The model provided significantly less information about a given ConSink cell’s spiking when using any of these positions rather than the identified sink position itself (Wilcoxon signed-rank test, all comparisons $p < 0.001$).”

11) If you take the small range of spatial positions represented by a place field and constrain the animal’s head direction because of task structure you might produce egocentric coding to *any* position in the arena.

Response: Yes, which is why it is important that the correction for directional bias takes into account the locations at which a given cell fired its spikes, as we have done, rather than correcting for bias averaged across the whole maze (see Methods, line 512-528).

12) Claims regarding density of ConSinks near the goal and their movement relating to changes to goal position are not well supported by the data presented. The study should include an analysis to verify that the reported distance metrics and their relative changes are statistically different than a random distribution of pairwise comparisons between all other platforms.

Response: We performed the analysis suggested by randomly picking pairs of platforms for goal 1 and goal 2, with the only stipulations being that goal 1 and goal 2 could not use the same platform, nor could they use the same platforms as were actually used in the session. For each random pair of platforms, we calculated the median distance of all ConSinks from either epoch to both Goal 1 and Goal 2, and calculated a scaled difference between them. We then calculated the same scaled differences from the real data, using the median values shown in manuscript Fig. 2b and d. We found that the difference in distances to Goal 1 and 2 was significant relative to the control distribution during the Goal 1 epoch (Extended Data Fig. 10a), but did not quite reach significance during the Goal 2 epoch (Extended Data Fig. 10b). However, we also asked what was the probability of simultaneously obtaining both the Goal 1 and Goal 2 epoch results; this turned out to be significant at alpha

level 0.001, supporting our interpretation that sinks were closest to Goal 1 during the Goal 1 epoch, and moved towards Goal 2 during the Goal 2 epoch (Extended Data Fig. 10c). Further, if we accept that our measurement of ConSink position is not perfectly accurate, we can ask whether the sinks have moved relatively close to the goal (i.e. to at least within the 6 surrounding platforms) by omitting the surrounding platforms from the random control distribution; in this case, the movement towards Goal 2 and its surrounding platforms is highly significant (panel d, $p < 0.001$).

This data is presented in Extended Data Fig. 10 (line 861).

Extended Data Figure 10 | ConSink location in the goal switch session is non-random. a, During the Goal 1 epoch, the difference in sink distance to Goal 2 relative to Goal 1 (normalized by distance to Goal 1) was greater than for random pairs of platforms ($p < 0.05$). b, During the Goal 2 epoch, the difference in sink distance to Goal 1 relative to Goal 2 (normalized by distance to Goal 2) did not quite reach significance compared to random pairs of platforms. c, However, the sum of the Goal 1 and Goal 2 epoch differences were highly significantly different from the values obtained from random platforms ($p < 0.001$), supporting the interpretation that ConSinks cluster around the goal. d, If we instead ask, during the Goal 2 epoch, whether the sinks were closer to the general vicinity of Goal 2 (i.e. Goal 2 and its 6 immediately surrounding platforms), accomplished by omitting these surrounding platforms from the random pairs distribution, then the difference from random platforms is highly significant ($p < 0.001$).

13) The study should include a substantially expanded analysis of reliability. How much do ConSink positions change from trial to trial when there is no goal or when the goal is stationary for an extended period of time.

Response: We're sorry, but unfortunately we don't think there are enough spikes fired, nor enough locations on the maze occupied, on individual trials to provide accurate estimates of the sink positions. We do have a comparison of the first and second halves of session 1 in Extended Data Fig. 13. We have now added plots showing explicitly the movement of sinks from the first half to the second half (Extended Data Fig. 13c).

We have added to the figure legend (line 892):

“c, Most ConSinks move towards the goal. Arrowheads refer to locations of ConSinks in (b), tails to ConSinks in (a).”

14) The fantail population analysis is underdeveloped and should be described in greater detail. Like other parts of this work one wonders if the distributions could be explained by behavioral biases of the animal. How much do the neural fantails resemble relative heading occupation?

Response: There is little resemblance between the fantails and the plots of occupancy, which are shown in Extended Data Fig. 17. In particular, during the initial wait period, it is quite clear that the animals are not oversampling the goal direction

(Extended Data Fig. 17a left panel), despite the fantail at this time point showing a goal-oriented distribution (manuscript Fig. 4d).

15) Raw data points for all boxplots please

Response: As far as we are aware, boxplots are currently viewed in the field as an acceptable way (along with dot plots) of presenting data, as opposed to bar graphs, which are banned at some journals. Note that all outliers are clearly shown in our figures, thus we hide nothing by using them; many of the plots are too small, and have values that are too concentrated to be able to display the data clearly with raw data points (e.g. Fig. 4, on error trials, where many values are 0).

16) L122 Fig. 1j instead of i?

Response: Panel i is the correct call out here. Panel j is showing that most ConSink cells have a preferred relative direction towards their corresponding sinks of around 0 degrees, but these sinks are not all at the goal. However, panel i shows that the population ConSinks, calculated using the platform-associated population vectors (see Methods line 570), are close to the goal. The text which initially referenced Fig. 1i and j was confusing, and we have clarified it (line 60):

“The ~~average of the~~ population vector fields (calculated by summing platform-associated heading vectors across cells) for each animal converged to a population ConSink close to the goal (Fig 1i) and the average heading direction preferred directions of ConSink place cells towards their sinks was in front of the animal, -19.2° , not significantly different from 0° (Fig 1j).”

17) Spatial information, by itself, is a poor metric for identifying place cells in that it simply reflects bimodalities in the firing rates across locations. A measure, such as coherence, that reflects the clustering of high and low firing rates should be used to complement the spatial information measure.

Response: We agree and have now calculated coherence (i.e. the correlation between a cell's rate map and a rate map created from the mean values of each bin's nearest neighbours, the 8 bins surrounding a bin of interest). To calculate a control distribution to test significance, we drew the bins for the neighbour rate map randomly (using only occupied bins). All ConSink cells had significant coherence.

We have added text (line 48):

“We recorded 266 CA1 place cells (defined as having significant spatial information¹⁵ and coherence¹⁶,”

18) One can imagine simply using a threshold for number of spikes emitted to pare down the dataset.

Response: We only calculated ConSinks using principal cells that fired at least 500 spikes during the corresponding epoch. All these cells had significant spatial information, so calculating SI or not makes no difference to the data as presented.

19) Why constrain the analyses to location-specific firing when two measures of orientation and distance are under consideration. Based on the analysis design, the work takes as an assumption the primacy of location in triggering firing of all CA1 neurons and should perhaps not.

Response: As per the response above, calculating SI did not affect the data as presented as all ConSink cells had significant SI. Our analysis didn't have any “in-built” assumptions about the primacy of location that would have prevented us from identifying relative direction cells that were not sensitive to location (and indeed, we identified a small number of cells, 9.1%, that were significant for relative direction but not sink distance or allocentric direction to sink in the Hardcastle analysis).

20) With respect to the inclusion/exclusion of spikes occurring during sharp wave ripple events, the criteria seem reasonable, but the authors should compare the tuning for both kinds of spikes. Why presume that the relevant moments are all associated with theta patterned activity? It seems at least plausible that ripple-related activity could be more meaningful given the results of so many Foster laboratory publications.

Response: We looked at the spikes that were initially excluded due to our combined criteria (increase in ripple power and pop'n rate, decrease in theta) across our entire population of ConSink and non-ConSink cells. Using only these spikes, no cells tested significant for ConSink tuning. This was not a huge surprise, as ripple activity is generally thought to reflect paths including locations away from the actual location of the animal. While the locations of these paths can be biased towards starting or ending at the animal's actual location, we don't know whether this bias is influenced at all by heading direction (i.e. must the animal assume the preferred heading of a given cell for it to be (re)activated within the ripple?). It is interesting to speculate on how ConSink firing might be organized within ripples, but we think this question is beyond the scope of the current study.

21) The authors should generate a quantification (perhaps using coherence or diffusion metrics) that

speaks to the concentration of high values in the maps of convergence resultant lengths. If ConSinks are to be taken very seriously, one would expect to see that a mapping convergence resultant lengths would look rather like the ratemap of a place cell.

Response: We investigated this using coherence, as suggested. Briefly, we constructed maps of MRL values; these were the MRL values corresponding to the vectors shown in the individual cell field vector plots (panels a-c, left and middle panels). We then calculated the correlation of the linearized MRL map with a linearized “neighbour” map; this “neighbour” map was constructed by averaging the MRL values directly surrounding each pixel in the MRL map. Of the 77 ConSink cells from Session 1, 24 had significantly positive correlations (panels a-c, right), while 3 had significantly negative correlations. Pooling the correlation coefficients across all 77 animals showed that the correlations were overwhelmingly positive (panel d; $p < 0.001$). We also ran a complementary analysis where we correlated MRLs with firing rates; the effect here was weaker, with 13 cells with significant positive correlations and 4 with negative correlations. Again, pooling the correlation coefficients showed the relationship between rate and MRL was positive (panel e; $p < 0.001$). Unlike coherence using a rate map, zero firing rate bins can't be used since they don't have a meaningful MRL value, which makes these measures somewhat noisier, particularly in cells whose firing covers a smaller area (e.g. panel c, Rat 3 TT27c3); this may contribute to the lower than expected significance in individual cells.

ADDITIONAL CHANGES

1. We found and corrected an error in our analysis code which corrected the fantail distribution by directional occupancy. This has changed the absolute firing rate values in Fig. 3k and l (previously Fig. 4k and l), but not the general shape of the distributions from which we drew our conclusions. Note that the fantails presented in Fig. 4d (formerly Fig. 5d) were not affected by this error.
2. In Figure 2f, the significance was incorrectly labelled ** $p < 0.001$. It has been corrected to * $p = 0.023$.
3. In Figure 4e, we had originally performed a Wilcoxon rank sum test, which is unpaired, because we had eliminated a small number of cells from the Error

choices data due to their lack of firing in any heading direction (5 during wait period 1, and 8 during wait period 2). Because the underlying data will be available in the online version of paper, to avoid confusion for the reader, we are now including these cells and performing the statistics using a Wilcoxon signed rank test, which is paired. This changes the p value for Wait Period 1 to $p = 0.006$; Wait Period 2 remains $p < 0.001$. The boxplots have been updated accordingly.

The text has been updated (line 393):

“Wilcoxon signed rank test, $p = 0.006$ (top), $p < 0.001$ (bottom).”

4. We have included a reference to the work of Edmund Rolls on spatial view cells in primates (line 31):

“Previous work has shown that cells in the hippocampal formation encode heading towards a goal and other locations in the bat¹², mouse⁷, primate¹³, and human¹⁴.”

Rolls, E. T. & O'Mara, S. M. View-responsive neurons in the primate hippocampal complex. *Hippocampus* 5, 409-424, doi:10.1002/hipo.450050504 (1995).

5. We have moved what was Figure 3 in the initial submission to Extended Data Figure 12, reducing the number of main figures from 5 to 4.
6. We have edited the manuscript in order to reduce the word count without sacrificing legibility, and it now stands at 193 words for the summary and 2467 words for the main text.

Reviewer Reports on the First Revision:

Referees' comments:

Referee #1 (Remarks to the Author):

I am satisfied by the response of the authors to the reviewer comments. This is an exciting and important paper using innovative techniques to demonstrate neurophysiological dynamics of place cells directly relevant to guiding navigation behavior.

Referee #2 (Remarks to the Author):

The authors have responded well to most of the concerns of the prior review, with the exception of Point 1:

1) Regarding Major point 1 of my prior review:

(a) The text (line 52) states that the firing was “best described” by vector fields. This implies that the firing was compared across different models or hypotheses in order to make a judgment that the vector fields “best describe” the data, which is why I conjectured that the authors were comparing the data against the allocentric model to support this statement. In the absence of any alternative hypothesis that was tested, the authors cannot use the term “best described”, but rather something like “firing patterns that could be described by vector fields” or similar.

(b) The much larger problem is that the authors’ new analyses in Extended Data Figure 7 do not address the fundamental concern I raised in the earlier review. The problem I raised is rather simple: It is an extremely biased comparison to test the value of a single measurement of one variable (allocentric head direction tuning) against the *best* measurement of another variable (egocentric head direction tuning). The authors calculate egocentric tuning by finding the consink that produces the maximal resultant vector length out of 986 (34 x 29) potential consinks (line 501). They compare this to the allocentric tuning curve, which by definition produces a single value. It is not unlikely that one of the consinks (especially a consink outside the maze) will produce an egocentric tuning curve for an allocentric head direction cell that happens to be slightly higher than the allocentric curve, because the egocentric tuning curve is biased to choose the best out of 986 measures. Unless I am misunderstanding, all that the authors have shown in Extended Fig 7 is that, regardless of the noise introduced into the system, the same bias in favor of egocentric > allocentric is present. There is little evidence in this paper that many of the so-called egocentric consink cells are not allocentric head direction tuned neurons. I am not saying that this is the case, but it may be true for many of the putative consink cells. Thus, the statement in lines 64-65 and the analysis of Extended Figure 7a may be flawed and provide a very misleading picture of the strength of allocentric over egocentric tuning for every cell.

2) The title is misleading. The authors have not shown that place cells use vector computations to navigate; they have shown correlations, not causations. The title should be changed to better reflect what has been shown here (similar to more reserved statements in the paper in response to my Point 15 from the prior review). For example, “Vectorial navigation properties of hippocampal place

cells” would be appropriate in that it does not state that the place cell use these properties for navigation, a finding that has not been directly demonstrated in this paper.

3) P. 58: It is still clear from Figure 1h that for Rat 2, the ConSinks were not densest around the goal. This sentence should read “they were densest around the goal for 2 of 3 animals” or similar

5) Fig. 2d: Is this all ConSink cells, or just those active only in Goal 2 (as was specified in Figure 2c)?

Referee #3 (Remarks to the Author):

The authors have undertaken a number of new control analyses that strengthen their case for the main claim of the manuscript, that consyncs (egocentric directional tuning to an individual maze location) are clustered over a space surrounding a goal target. The activity of neurons exhibiting this property are also proposed to exhibit varying degrees of tuning to allocentric position and distance to a goal. The tuning properties of these neurons are proposed as “solutions” to the navigational problem present during the “honeycomb” task of choosing a next location (among two) that moves the animal toward the eventual goal.

The nature of the task is certainly interesting and provides an open-field task for rodents that demands navigation to a hidden goal defined by distal visual cues. The task itself, however, has been published and the authors overstate the extent to which it mirrors real-world navigational problems and the extent to which it is novel in involving a series of choice points (see, for example, data from Ainge/Dudchenko or Redish). Nevertheless, the attempt to examine in detail how hippocampal ensembles alone might solve the task is commendable and the work begins to pick apart the critical interface between directional and spatial tuning.

A continuing weakness of the present work concerns the number of neurons included for analysis and the number of animals utilized. Only a small percentage of the neurons exhibit the “consync” property which raises the question as to what the remaining hippocampal neurons contribute to task performance. Furthermore, the criterion for being considered a “consync” neuron is relatively weak (exhibiting tuning beyond the 95th percentile of a completely random distribution). A main claim is that consyncs are clustered near goal sites, but the clustering is just not impressive even if it is statistically beyond chance by a number of new control analyses. In the end, it is hard to believe that the tight tuning to space in hippocampal pyramidal cells and to head orientation in neurons across many regions of the brain yields such noisy tuning. It is much easier to conceive of the weak directional tuning bias of these cells as a non-spatial determinant of in-field firing than it is to now think of the hippocampus as robustly encoding combinations of egocentric orientation to a goal and allocentric location. The work is less than definitive, but could conceivably be more convincing were much more data considered.

Notably, the interpretation of the findings rather assumes that the directional signal leads to a

choice among the next two possible maze locations. In fact, the nature of the task can also be thought of as involving a left versus right choice that has some similarity to the left-right choice-making used in a multitude of T-maze experiments. In many and most such experiments, hippocampal place cells are strongly modulated by the upcoming choice in a dynamic fashion (see Cell paper from the Frank laboratory). Thus, the directional tuning seen in the present work could actually be secondary to the deliberation of left-right choice rather than a contributor to the choice itself. This is admittedly a subtle point, but the authors should acknowledge their inability to evidence a direction-first, action-second order of operation. It is quite possible that the hippocampal activity is modulated by choice rather than providing a directional signal upon which a choice could be made by a downstream recipient of hippocampal output. Notably, this is consistent with the enhanced consync tuning during the time period leading up to actual choice as compared to the tuning in the delay period before the choice locations are presented.

Author Rebuttals to First Revision:

Referees' comments:

Referee #1 (Remarks to the Author):

I am satisfied by the response of the authors to the reviewer comments. This is an exciting and important paper using innovative techniques to demonstrate neurophysiological dynamics of place cells directly relevant to guiding navigation behavior.

- We thank the Referee for his/her supportive comments.

Referee #2 (Remarks to the Author):

The authors have responded well to most of the concerns of the prior review, with the exception of Point 1:

1) Regarding Major point 1 of my prior review:

(a) The text (line 52) states that the firing was “best described” by vector fields. This implies that the firing was compared across different models or hypotheses in order to make a judgment that the vector fields “best describe” the data, which is why I conjectured that the authors were comparing the data against the allocentric model to support this statement. In the absence of any alternative hypothesis that was tested, the authors cannot use the term “best described”, but rather something like “firing patterns that could be described by vector fields” or similar.

- “best described” has been changed to “well described “ (Line 54).

(b) The much larger problem is that the authors' new analyses in Extended Data Figure 7 do not address the fundamental concern I raised in the earlier review. The problem I raised is rather simple: It is an extremely biased comparison to test the value of a single measurement of one variable (allocentric head direction tuning) against the *best* measurement of another variable (egocentric head direction tuning). The authors calculate egocentric tuning by finding the consink that produces the maximal resultant vector length out of 986 (34 x 29) potential consinks (line 501).

They compare this to the allocentric tuning curve, which by definition produces a single value. It is not unlikely that one of the consinks (especially a consink outside the maze) will produce an egocentric tuning curve for an allocentric head direction cell that happens to be slightly higher than the allocentric curve, because the egocentric tuning curve is biased to choose the best out of 986 measures. Unless I am misunderstanding, all that the authors have shown in Extended Fig 7 is that, regardless of the noise introduced into the system, the same bias in favor of egocentric > allocentric is present. There is little evidence in this paper that many of the so-called egocentric consink cells are not allocentric head direction tuned neurons. I am not saying that this is the case, but it may be true for many of the putative consink cells. Thus, the statement in lines 64-65 and the analysis of Extended Figure 7a may be flawed and provide a very misleading picture of the strength of allocentric over egocentric tuning for every cell.

- The referee is unconvinced by the analysis we presented in Extended Data Figure 7 (now Extended Data Fig. 8) in which, based on his/her original conjecture that the consinks were noisy HD cells, we showed that the addition of noise to putative allocentric head direction cells could not account for the data. Their first major objection is that because we compare the best of many measures of egocentric tuning to a single measure of allocentric tuning, we are likely to produce higher measures of egocentric coding. They state that our new analysis simply confirms this. We disagree. Instead, this analysis shows that a pure HD cell is unlikely to produce an egocentric MRL greater than its allocentric MRL. As we stated, this only occurs at very high levels of noise in the HD signal. Secondly, even at these higher levels of noise, the increase in egocentric vs. allocentric MRL is much smaller than we observe in our data.
- The referee is correct that it is more difficult to distinguish egocentric from allocentric tuning to points that lie off the maze. As shown in new panel Extended Data Fig. 8f, this is because the sampling of allocentric direction when the animal is oriented in the cell's preferred egocentric direction is much

less broad when sinks are off the maze. Unfortunately, there is little more we can do, besides the previous analysis modelling pure HD cells with noise, to verify that these ConSinks at off-maze positions are in fact true ConSinks. However, the relationship between sink position and sampling of egocentric direction provides an opportunity to further test the broader, and we think more important, question of whether the overall population of ConSink cells we have identified are truly tuned to egocentric direction. Referring again to Extended Data Fig. 8f, it is clear that egocentric tuned cells will appear more allocentric as the sink positions move further from the maze. Calculating the difference between egocentric and allocentric MRLs and plotting this against the distance of the sink from the maze centre shows that this is the case, and that the relationship is highly significant (Extended Data Fig. 8g-i).

- Given these data, the referee might adopt the position that some of the ConSinks beyond the perimeter are actually head direction cells and that although they only represent a small percentage of the overall population of ConSinks, they in fact are responsible for all of effects we report. To address the referee's concerns, we have looked at this argument in two ways, 1) by removing cells with ConSinks beyond the maze perimeter, and 2) removing those cells whose egocentric-allocentric MRLs are most similar.

The ConSinks beyond the perimeter represent 22% of the total in rat 1 (session 1: 3/21; session 2 goal 1: 8/47, goal 2: 9/25), 34% of the total in rat 2 (session 1: 10/27; session 2 goal 1: 3/14, goal 2: 8/21), 22% in rat 3 (session 1: 7/29; session 2 goal 1: 4/23, goal 2: 4/16), 20% in Rat 4 (session 1: 8/30; session 2 goal 1: 4/25, goal 2: 3/19), and 49% in Rat 5 (session 1: 17/35). The

effect of removing these ConSinks from the calculations of the average ConSink and the average vector fields is shown in Extended Data Fig. 8 j and k; it is clear that removing the cells with sinks off the maze has little effect on the overall results.

We repeated this analysis removing cells whose egocentric-allocentric MRL differences (calculated as $(\text{egoMRL} - \text{alloMRL})/(\text{egoMRL} + \text{alloMRL})$) were less than 0.05 (29% of the total in rat 1 (session 1: 6/21; session 2 goal 1: 13/47, goal 2: 8/25), 29% of the total in rat 2 (session 1: 10/27; session 2 goal 1: 3/14, goal 2: 5/21), 21% in rat 3 (session 1: 10/29; session 2 goal 1: 3/23, goal 2: 1/16), 14% in Rat 4 (session 1: 7/30; session 2 goal 1: 1/25, goal 2: 2/19), and 46% in Rat 5 (session 1: 16/35). The effect of removing these is shown in Extended Data Fig. 8l and m, and again there is little change from the original results. We hope the referee will agree that it is clear from this figure that the effects reported in the paper are still robust after the exclusion of these quasi-head direction cells. We have added the following sentence to the text “ConSink tuning was stronger in every ConSink cell than allocentric head-direction tuning and removing either putative allocentric candidates or cells with ConSinks beyond the maze perimeter where ConSink cells might be expected to appear most allocentric did not substantially alter the results (Extended Data Fig. 8j-m).” Line 68.

2) The title is misleading. The authors have not shown that place cells use vector computations to navigate; they have shown correlations, not causations. The title should be changed to better reflect what has been shown here (similar to more reserved statements in the paper in response to my Point 15 from the prior review). For example, “Vectorial navigation properties of hippocampal place cells” would be appropriate in that it does not state that the place cell use these properties for navigation, a finding that has not been directly demonstrated in this paper.

- We have changed the title to “**Hippocampal place cells exhibit goal-oriented vector fields during flexible navigation**”

3) P. 58: It is still clear from Figure 1h that for Rat 2, the ConSinks were not densest around the goal. This sentence should read “they were densest around the goal for 2 of 3 animals” or similar.

- We have changed the sentence to “While ConSinks were scattered around the environment both on and off the maze, they were densest around the goal in 4 of 5 animals (Fig. 1h, Extended Data Fig. 6a) and moved closer to the goal in the second half of the session (Extended Data Fig. 7); this clustering was not due to the positions of their place fields (Extended Data Fig. 6d-h).”

Line 59

5) Fig. 2d: Is this all ConSink cells, or just those active only in Goal 2 (as was specified in Figure 2c)?

- Just those active in Goal 2 (and panel b is just those active in Goal 1). This has been clarified in the figure legend.
 - **d, ConSink population active during Goal 2 was closer to Goal 2 than Goal 1.** Line 371

Referee #3 (Remarks to the Author):

The authors have undertaken a number of new control analyses that strengthen their case for the main claim of the manuscript, that consyns (egocentric directional tuning to an individual maze location) are clustered over a space surrounding a goal target. The activity of neurons exhibiting this property are also proposed to exhibit varying degrees of tuning to allocentric position and distance to a goal. The tuning properties of these neurons are proposed as “solutions” to the navigational problem present during the “honeycomb” task of choosing a next location (among two) that moves the animal toward the eventual goal.

The nature of the task is certainly interesting and provides an open-field task for rodents that demands navigation to a hidden goal defined by distal visual cues. The task itself, however, has been published and the authors overstate the extent to

which it mirrors real-world navigational problems and the extent to which it is novel in involving a series of choice points (see, for example, data from Ainge/Dudchenko or Redish).

- Referee 3 complains that we overstate the extent to which the honeycomb maze mirrors real-world navigational problems and questions its novelty in terms of providing a series of choice points. We would argue that it was not designed to mirror all real-world navigations and that its novelty resides in, but also goes well beyond, the number of choice points. The task was designed within the conceptual framework of the cognitive map theory which set out the different ways in which navigational problems could be solved and assigned a very specific role to the hippocampus within that framework. In brief, the theory posits that an animal can navigate using 1) a guidance which it approaches, follows or maintains in a particular part of its egocentric framework, 2) a route which consists of a sequence of turns in response to particular stimuli eg those at a choice point, or finally, 3) a more flexible mapping strategy which enables the animal to navigate to a particular goal from any location and direction, and to cope with obstacles by generating alternative novel routes. It is only this last navigational strategy which is postulated to depend on the hippocampus and which the honeycomb maze navigational task was designed to test. Importantly, this task cannot be solved by either of the two non-hippocampal strategies because, in common with the Morris water maze, there is no beacon either at the goal or elsewhere in the environment which can be consistently approached or oriented relative to, and there are no behavioural sequences which will be consistently successful because the animal is started from a different location and must use different sequences of turns to reach the goal on each trial. In fact, the start locations

and the sequence of required choices are such that there is almost never a behavioural sequence which is repeated across the trials of a day's testing. Because the animal must approach the goal from different directions, a simple uniform directional strategy is also ruled out. The drawbacks of the Morris water maze task are that it is very hard to score since there are a very large number of choices and directions which the animal can move in at any location and once the animal has learned the location of the goal it tends to limit its behaviour to movements in that direction ruling out an assessment of important aspects of place cell firing such as their directionality at different points within the place field. Comparison with the tasks mentioned by the referee highlights these advantages. The tasks used by Ainge/Dudchenko, Redish, and Frank all involve the animals running along linear tracks which constrain it to move in one of two directions at any point and to choose between one of two directions at choice points. In the Ainge and Frank tasks there is more than one goal further complicating the search for a goal representation. An added advantage of the honeycomb maze task is that the animal is confined for a short period between each choice affording the opportunity to examine the place cell representation at that location before the animal knows what choices it will be given there. We will return to this point in our response to a subsequent speculation by this referee about the relationship between place cell activity and the animal's subsequent choice. Finally, in behaviour reminiscent of the more restricted vicarious trial and error behaviour exhibited on some trials at the single choice point during T-maze learning, the animal tends to sample all of the possible directions during each platform confinement before it knows the available choices. It is this behaviour

which enables us to record the firing activity in different directions resulting in the Fantail representation. In summary, we did not claim that the honeycomb maze mirrors all real-world navigational problems but only that it is a powerful behavioural tool for examining hippocampal activity during hippocampal-dependent cognitive map-based flexible navigation.

A continuing weakness of the present work concerns the number of neurons included for analysis and the number of animals utilized.

- After discussions with the editor, we implanted and collected data from 2 additional animals, repeating our earlier experiments. We replicated our findings from the original 3 animals, and have added the data to the manuscript. One of the 2 new animals wasn't able to learn the goal switch in a single session, so was left out of the main analysis presented in Fig. 2 and associated Extended Data Figures. However, we were able to train this animal to the second goal over a number of days, allowing us replicate the main findings from the goal switch sessions in the other animals, specifically that the distribution of ConSinks moves towards the new goal, as does the population field vector; this data is presented across Extended Data Figures 9, 11 and 12.
- The addition of this new data did not render any previously significant analyses in the main figures non-significant, or vice versa. There were 2 relatively minor changes in the Extended Data Figures, however:
 - Extended Data Fig. 10f (formerly Extended Data Fig. 9f): Previously, we found in the Goal Switch sessions that cells with significant sinks to both Goal 1 and Goal 2 had preferred relative directions to the 2 sinks

that were more similar than expected by chance. This effect no longer reaches significance. Note that while this may be of interest to readers, we would not classify this as a central finding of our study in the sense that it was not previously, nor currently, mentioned in the text due to space constraints.

- Extended Data Fig. 11d (formerly Extended Data Fig. 10b): Previously, this analysis from the Goal Switch sessions examining the significance of the difference in sink distance to Goal 1 and Goal 2 did not quite reach significance, necessitating a slightly more involved analysis (shown in former panels c and d). This analysis now reaches significance, allowing us to remove the extra panels.

Only a small percentage of the neurons exhibit the “consync” property which raises the question as to what the remaining hippocampal neurons contribute to task performance.

- We disagree that the percentage of significant cells is “small”, but have to admit that we’re not sure how big or small percentages are defined. It seems that the reviewer is focusing on the session 1 data (31% significant cells), when the goal switch data finds that 45% of cells are significant in at least one of the goal epochs; while it doesn’t seem completely unreasonable to call 31% small, we think it is stretching the definition at 45%.

We agree that it is reasonable to ask what the remaining population of inactive cells might contribute. The referee will be aware that the number of place cells active in any given environment is variable, ranging from 25% to 90% depending on the enclosure and the task. Studies which have recorded

from the same hippocampal cells in multiple environments (Thompson and Best 1989, Alme et al. 2014) suggest that all hippocampal pyramidal cells will eventually have a place field in at least one environment if enough environments are tested. By analogy, we would assume that all hippocampal pyramidal cells would demonstrate ConSink coding in a navigation task in at least one environment, if enough were tested. Support for this hypothesis comes from our demonstration in the goal switch experiment that many of the cells expressing egocentric tuning during the first goal did not during the second, when many new cells began to participate in the ConSink representation. Presumably these cells were simply waiting for the right goal location to become active.

Furthermore, the criterion for being considered a “consync” neuron is relatively weak (exhibiting tuning beyond the 95th percentile of a completely random distribution). A main claim is that consyncs are clustered near goal sites, but the clustering is just not impressive even if it is statistically beyond chance by a number of new control analyses. In the end, it is hard to believe that the tight tuning to space in hippocampal pyramidal cells and to head orientation in neurons across many regions of the brain yields such noisy tuning. It is much easier to conceive of the weak directional tuning bias of these cells as a non-spatial determinant of in-field firing than it is to now think of the hippocampus as robustly encoding combinations of egocentric orientation to a goal and allocentric location. The work is less than definitive, but could conceivably be more convincing were much more data considered.

- We disagree that the tuning is weak and noisy and do not know how to respond to the subjective criticism of our use of the standard measure for significance. What the referee sees subjectively as noisy tuning we would argue is part of the mechanism which allows the animal to compare sub-optimal directions to each other, something narrow tuning would not allow. We do not know how a weak non-spatial directional tuning bias of place cells

proposed by the referee would lead to a representation which would support flexible navigation in the way that our interpretation of the data does or what other purpose it could serve, and so feel unable to respond to this conjecture.

Notably, the interpretation of the findings rather assumes that the directional signal leads to a choice among the next two possible maze locations. In fact, the nature of the task can also be thought of as involving a left versus right choice that has some similarity to the left-right choice-making used in a multitude of T-maze experiments. In many and most such experiments, hippocampal place cells are strongly modulated by the upcoming choice in a dynamic fashion (see Cell paper from the Frank laboratory). Thus, the directional tuning seen in the present work could actually be secondary to the deliberation of left-right choice rather than a contributor to the choice itself. This is admittedly a subtle point, but the authors should acknowledge their inability to evidence a direction-first, action-second order of operation. It is quite possible that the hippocampal activity is modulated by choice rather than providing a directional signal upon which a choice could be made by a downstream recipient of hippocampal output. Notably, this is consistent with the enhanced consync tuning during the time period leading up to actual choice as compared to the tuning in the delay period before the choice locations are presented.

- The referee claims that the nature of the present behavioural task can also be thought of as involving a left vs right choice that has some similarity to the left-right choice-making used in a multitude of T-maze experiments. Furthermore, his/her interpretation of the firing of place cells prior to the choice on these simple binary choice tasks is that the animal is deliberating on the left-right choice to be made. At the risk of labouring several of the points made in response to an earlier comment, we would reject this interpretation on the following grounds. In the T-maze task, the animal is repeatedly faced with the same binary choice at the same location in space. It very quickly learns that before it reaches the choice point it will be faced with that binary option at the choice point. Therefore, it is plausible to postulate that it can entertain that limited choice prior to arriving at the choice point and that hippocampal place cells might represent those mental considerations. In contrast, in the present

task the animal is faced with the possibility of greater than 500 binary choices across 60 locations, the sequence of which differs on each traversal.

Furthermore, the choices cannot be simply characterized as left versus right as they can on the T-maze where the animal approaches the two possibilities from the same direction and thus they lie on one half of the body axis or the other. No simple orientating axis exists on the platforms of the honeycomb maze given the animal's frequent rotations while it is waiting on the platform for the next choice platforms to be presented. Even were the animal able to maintain a memory of the orientation axis given by its trajectory from the previous choice, many of the choices involve platforms on the same side of this axis (i.e. left 60° vs left 120°). In sum, we do not think it feasible to represent the animal's cognitive processes while it waits for the next platforms to be raised to consist of the consideration of a very large number of potential binary choices in the same way that it is possible to do so on a simple T-maze task. Rather, as our fantail model suggests, it is simpler to represent the animal's cognitive processes as involving an ordered assessment of the possible directions of movement in terms of their angle to the goal direction. We are open to the possibility that a similar mechanism is operating at the choice point in the T-maze but the limitations of that task and apparatus do not allow it to be fully explored.

- We also do not agree that the increased goal-oriented tuning of the ConSink in the second part of the delay when platforms are rising provides definitive evidence for a role in the actual upcoming choices on the ConSink configuration because it might merely be due to the passage of time and since we did not significantly vary the length of time before the platforms started to

rise, we cannot rule this out as a possibility. However we cannot see how these data could be used to infer the consideration of a large number of potential binomial choices during the waiting period before the platform begins to rise and the animal knows which choices will actually be available to it.

* Nature Research's authors website (<https://www.nature.com/authors>) contains information about and links to policies and resources.

Our flexible approach during the COVID-19 pandemic

If you need more time at any stage of the peer-review process, please do let us know. While our systems will continue to remind you of the original timelines, we aim to be as flexible as possible during the current pandemic. We are committed to processing manuscripts related to COVID-19 with our highest priority, and endeavour to expedite the peer-review process for COVID-19 submissions as much as is possible. Thank you for your understanding.

This email has been sent through the Springer Nature Manuscript Tracking System NY-610A-SN&MTS

Confidentiality Statement:

This e-mail is confidential and subject to copyright. Any unauthorised use or disclosure of its contents is prohibited. If you have received this email in error please notify our Manuscript Tracking System Helpdesk team at <http://platformsupport.nature.com>.

Details of the confidentiality and pre-publicity policy may be found here <http://www.nature.com/authors/policies/confidentiality.html>

Privacy Policy | Update Profile

DISCLAIMER: This e-mail is confidential and should not be used by anyone who is not the original intended recipient. If you have received this e-mail in error please inform the sender and delete it from your mailbox or any other storage mechanism. Springer Nature America, Inc. does not accept liability for any statements made which are clearly the sender's own and not expressly made on behalf of Springer Nature America, Inc. or one of their agents.

Please note that neither Springer Nature America, Inc. or any of its agents accept any responsibility for viruses that may be contained in this e-mail or its attachments and it is your responsibility to scan the e-mail and attachments (if any).

Reviewer Reports on the Second Revision:

Referees' comments:

Referee #2 (Remarks to the Author):

The new analyses that the authors performed in Extended Data Figure 8 address my major concerns about the inherent bias in the analysis to favor egocentric tuning over allocentric tuning, and the results of these analyses argue convincingly that a good fraction of the cells with Consink tuning are egocentric and not allocentric. The authors and I will have to agree to disagree on whether the simulations of adding noise really address the problem I raised, as I am still not convinced about that part, but the new analyses are satisfactory.

I have no further concerns.

Referee #3 (Remarks to the Author):

The authors have added data from two subjects to the manuscript's results. This is a welcome addition, but I still find the evidence for ConSinks to be relatively weak. Are neurons without ConSinks truly just silent neurons as the authors suggest?

The spatial dispersion of ConSinks is still troubling. In addition, I agree with referee 2's assessment that the comparison of the best egocentric tuning versus the only allocentric tuning represents a bias. I don't find the approach of adding noise to be terribly definitive or direct in addressing that criticism.

It is difficult to understand the authors' rather crass response ("Referee 3 complains...") to the issue raised over the description of the task and its relevance to the field, especially as the subsequent discussion of the point constrains the topic to only 3 possible navigational strategies outlined in a theory that is over 4 decades old.

Author Rebuttals to Second Revision:

Response to Referees

Referee #2 (Remarks to the Author):

The new analyses that the authors performed in Extended Data Figure 8 address my major concerns about the inherent bias in the analysis to favor egocentric tuning over allocentric tuning, and the results of these analyses argue convincingly that a good fraction of the cells with ConSink tuning are egocentric and not allocentric. The authors and I will have to agree to disagree on whether the simulations of adding noise really address the problem I raised, as I am still not convinced about that part, but the new analyses are satisfactory.

I have no further concerns.

Referee #3 (Remarks to the Author):

The authors have added data from two subjects to the manuscript's results. This is a welcome addition, but I still find the evidence for ConSinks to be relatively weak. Are neurons without ConSinks truly just silent neurons as the authors suggest?

- It was not our intention to suggest that cells without ConSinks were silent. In fact, the very first sentence of the results section (line 48) highlights the larger population of place cells we recorded, defined by their significant spatial information and coherence, indicating clearly that they were not silent cells (which by definition would not have significant spatial information or coherence due to their silence). Similarly, the second sentence of the discussion specifically mentions that the larger CA1 population, which includes non-ConSink cells, "*provides information about the animal's current location*" (line 197).
- To highlight the non-silent nature of these cells, we have changed the first sentence of the discussion (line 194):
 - During navigation on the honeycomb maze, the firing patterns of a subset of CA1 hippocampal pyramidal place cells are organized as vector fields oriented around a set of featureless environmental locations called ConSinks.

The spatial dispersion of ConSinks is still troubling.

- We think the point that the referee is making here is that based on our population analysis, the ConSinks should all be located at the goal location, and any deviation from this within individual cells represents noise in the

encoding of the sink positions. We don't think this interpretation is correct. Rather, we think that many ConSinks are actually located away from the goals, potentially allowing navigation to non-goal locations, which might be useful under certain conditions (e.g. when there is an obstacle in the direct path to goal). We think the coding scheme can be viewed as a hierarchy, where, at the top, is the population coding scheme which allows the animal to navigate directly to the goal, and underneath, the individual cell coding which allows navigation to other locations when necessary and further enhancing the flexibility of navigation allowed by ConSink cells.

The first paragraph of the discussion clearly acknowledges that the sinks are distributed around the environment (line 203). Further, we discuss the continued movement of sinks towards the goal within a session (line 208), which indicates that ConSink positioning away from the goal isn't simply due to noise but rather to the animal's experience with the task and navigating to the goal.

We don't think there is anything we can add that will satisfy Referee 3 on this point.

In addition, I agree with referee 2's assessment that the comparison of the best egocentric tuning versus the only allocentric tuning represents a bias. I don't find the approach of adding noise to be terribly definitive or direct in addressing that criticism.

- As reviewer 2 notes, the analysis we added to Extended Data Fig. 8 (now Extended Data Figures 9 and 10) addresses this point, and we therefore have not added further text addressing this.

It is difficult to understand the authors' rather crass response ("Referee 3 complains...") to the issue raised over the description of the task and its relevance to the field, especially as the subsequent discussion of the point constrains the topic to only 3 possible navigational strategies outlined in a theory that is over 4 decades old.

- It is widely agreed in the field that for spatial navigation, the hippocampus is specifically required when a flexible mapping strategy is required (as opposed to navigation to a beacon or landmark or requiring repeated sequences of right and left turns at familiar choice points). We think that the relevance and advantages of our task, i.e. the large number of unique choices both in terms

of the choice points themselves as well as the unique combinations of choices available on repeated visits to the same choice point, will be obvious to most researchers in the field even without the discussion of these points already included in the text, and is without precedent in the field. We don't think there is anything we can add that will satisfy Referee 3 on this point.